



**Effect of crustose lichen (*Ochrolecia frigida*) on soil CO$_2$**
**efflux in a sphagnum moss community over western Alaska**
**tundra**
**Yongwon Kim[1]\*, Sang-Jong Park[2], and Bang-Yong Lee[2]\*\***
[1]{International Arctic Research Center, University of Alaska Fairbanks, AK 99775-7335,
USA}
[2]{Division of Polar Climate Scienes, Korea Polar Research Institute (KOPRI), Incheon 21990,
Republic of Korea}
\* Correspondence to: Yongwon Kim (kimywjp@gmail.com).
\*\* Corresponding co-author: Bang-Yong Lee (bylee@kopri.re.kr).
Key words: Crustose lichen, CO$_2$ efflux, Soil temperature, Tundra, Sphagnum moss, Alaska
Running title: Effect of crustose lichen on soil CO$_2$ efflux





## 1  Abstract

Soil $CO_2$ efflux-measurements represent an important component for estimating an annual carbon
budget in response to changes in increasing air temperature, degradation of permafrost, and
snow-covered extents in the Subarctic and Arctic. However, it is not widely known what is the
effect of curstose lichen (*Ochrolecia frigida*) infected sphagnum moss on soil $CO_2$ emission,
despite the significant ecological function of sphagnum, and how lichen gradually causes the
withering to death of intact sphagnum moss. Here, continuous soil $CO_2$ efflux measurements by a
forced diffusion (FD) chamber were investigated for intact and crustose lichen sphagnum moss
covering over a tundra ecosystem of western Alaska during the growing seasons of 2015 and
2016. We found that $CO_2$ efflux in crustose lichen during the growing season of 2016 was 14 %
higher than in healthy sphagnum moss community, suggesting that temperature and soil moisture
are invaluable drivers for stimulating soil $CO_2$ efflux, regardless of the restraining functions of
soil moisture over emitting soil carbon. Soil moisture does not influence soil $CO_2$ emission in
crustose lichen, reflecting a limit of ecological and thermal functions relative to intact sphagnum
moss. During the growing season of 2015, there is no significant difference between soil $CO_2$
effluxes in intact and crustose lichen sphagnum moss patches, based on a one-way ANOVA at
the 95 % confidence level ($p < 0.05$). Considering annual soil $CO_2$ effluxes simulated by
temperature, as well as monitoring of snow depth by time-lapsed camera, average snow-covered
and snow-free $CO_2$ contributions to annual carbon budgets correspond to 28.4 % and 71.6 % in
intact sphagnum moss cover, and 25.0 % and 75.0 % in a crustose lichen sphagnum moss colony,
respectively. Therefore our findings demonstrate that soil $CO_2$ emissions in the crustose
lichen-infected sphagnum moss community would be steadily stimulated by a widespread
outbreak of airborne plants over intact sphagnum moss, and by a rapid degradation of permafrost
in response to drastic changes in climate and environment in the Subarctic and Arctic.



## 1 Introduction

Soil carbon dioxide ($CO_2$) efflux, produced through the decomposition of soil organic carbon and roots, signifies the second largest terrestrial carbon source on both time and space scales (Raich and Schlesinger, 1992; Schlesinger and Andrews, 2000; Bond-Lamberty and Thomson, 2010). This efflux is susceptible to increasing air temperature (ACIA, 2004; AMAP, 2011), the degradation of permafrost (Schuur et al., 2009; Jensen et al., 2014; Lawrence et al., 2015; Natali et al., 2015), changing snow cover extent (AMAP, 2011), and the expansion of the shrub community (Sturm et al., 2005; Bhatt et al., 2013). All of this suggests an alteration of the terrestrial carbon cycle in response to drastic changes in climate and environment in the Arctic (ACIA, 2004; AMAP, 2011). These changes affect the high-latitude terrestrial carbon cycle and budget, via changes in vegetation productivity (Euskirchen et al., 2006; Barr et al., 2007; Bhatt et al., 2013), decomposition of soil organic matter (Piao et al., 2008; Wu et al., 2012), and the degradation of permafrost (Schuur et al., 2009; Jensen et al., 2014; Lawrence et al., 2015; Natali et al., 2015). Of the changes documented in the Arctic, an increase in temperature is most important, as it drives positive feedbacks on regional and pan-Arctic scales (Chapin et al., 2000; ACIA, 2004). Soil carbon dynamics in tundra and boreal forest ecosystems represent strong temperature sensitivity, a factor characterized by $Q_{10}$ value, which describes an increase in respiratory rate with a given 10 °C temperature change (Xu and Qi, 2001; Davidson and Janssens, 2006; Bond-Lamberty and Thomson, 2010; Mahecha et al., 2010; Kim et al., 2013; 2014a; 2014b; 2016; Kim, 2014). Bond-Lamberty and Thomson (2010) estimated a global soil respiration rate of $98 \pm 12$ GtC ($1 GtC = 10^{15}$ gC), indicating an increase of 0.1 GtC year$^{-1}$ over two decades. This rate of increase suggests a $CO_2$ emission response factor of 1.5 compared to air temperature, which is consistent with enhanced soil $CO_2$ emission response to a warming global climate.

Sphagnum moss (*Sphagnum* spp.) is widely distributed over the permafrost regions of the Subarctic and Arctic, and the thermal insulative capacity and preservation of permafrost is strongly influenced by the water content of the moss layer (Yoshikawa et al., 2004). Living sphagnum mosses have impressive water holding potential, with a number of species able to hold

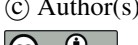



twenty or more times as much water as their dry weight (Turetsky et al., 2010). Sphagnum moss
habitats store large amounts of carbon, which helps reduce global warming (Fraser and Keddy,
2005). Nevertheless, crustose lichen (*Ochrolecia frigida*) infects the living sphagnum moss
community through airborne spread and finally causing the withering to death of healthy
sphagnum moss. Lichen is a composite organism that arises from algae, with cyanobacteria
living among filaments of multiple fungi species in a mutualistic relationship (Vitt et al., 1988;
Hasselbach and Neitlich, 1998; Spribille et al., 2016). Lichens may have tiny, leafless branches
(fruticose), a flat leaf-like structure (foliose), flakes that lie on the surface like a peeling plant
(crustose), a powder-like appearance (leprose), or other growth forms (Hasselbach and Neitlich,
1998; USDA, 2006). Lichens do not have roots that absorb water and nutrients as plants do, but
like plants, they produce their own nutrition by photosynthesis in foliose and fruticose forms
(Hahn et al., 1993; Otto et al., 1996; Hasselbach and Neitlich, 1998; Inoue et al., 2014). Most
lichens produce abundant sexual structures and appear to disperse only by sexual spores
(Murtagh et al., 2000). The crustose lichens *Graphisscripta parella* and *Ochrolecia frigida*
reproduce sexually by self-fertilization (i.e., they are homothallic). This breeding system enables
successful reproduction in harsh environments (Murtagh et al., 2000). However, it is not well
known what is the influence of crustose lichen-infected sphagnum moss cover, which is
commonly distributed on several moss species and peats in the high Arctic (Gary Laursen; personal
communication). The crustose lichen *O. frigida* is a sorediate Arctic lichen that grows on plant
materials, displays pink ascoma discs (Hasselbach and Neitlich, 1998), and shows high
adaptation for light reflectance (Hahn et al., 1993; Otto et al., 1996). Thus, if crustose lichens
invade over sphagnum moss cover, the moss could wither and die, losing its preservation of
permafrost. Here we investigated the difference in soil carbon emission from healthy and
crustose lichen-infected sphagnum communities in a tundra ecosystem during the growing
season.
Temperature and soil moisture are the most significant parameters for governing soil $CO_2$ efflux
across the tundra and boreal forest ecosystems of the Subarctic and Arctic (Lloyd and Taylor,
1994; Davidson et al., 1998; Davidson and Janssens, 2006; Rayment and Jarvis, 2000; Oberbauer
et al., 2007; Kim et al., 2007; 2013; 2014a; 2014b; 2016; Jansen et al., 2014; Kim, 2014;



Euskirchen et al., 2017); further, these environmental parameters must be efficiently validated for terrestrial ecosystem process-based models (e.g., Land Surface Models), for the assessment of carbon balance and budgets on regional and global scales. Consistent exertions are needed to evaluate these environmental parameters modulating soil $CO_2$ efflux in the sphagnum moss community of the tundra ecosystem during the growing season. Euskirchen et al. (2017) found that increases in air temperature and soil temperature at soil depths may have triggered a new trajectory of $CO_2$ release from 2008 to 2015, which would be a significant feedback toward further warming if it is representative of large areas of the Arctic.

The purposes of this study are to 1) determine the environmental drivers resolving soil $CO_2$ emissions in intact and crustose lichen-infected sphagnum moss regimes of the tundra ecosystem in western Alaska; 2) estimate soil $CO_2$ emission in sphagnum moss communities by continuous forced diffusion (FD) chamber system during the growing seasons of 2015 and 2016; and 3) assess the contributions from seasonally snow-covered- and snow-free-period carbon toward the simulated annual carbon budget, based on in-situ temperature and snow depth.

## 2  Materials and Methods

### 2.1 Sampling Descriptions and Methods

The crustose lichen (*Ochrolecia frigida*) strongly adheres to a substrate such as sphagnum moss, making separation from the substrate impossible without destruction. Generally, crustose lichens that cling to soil, rock, and tree bark can be found in a wide range of areas. In this study, we found several crustose lichen colonies on sphagnum moss in the tundra ecosystem of western Alaska (Supplementary Figure S1). Crustose lichen eventually causes a withering to death of sphagnum moss that has preserved discontinuous permafrost from degradation, due to protection from water evaporation (Yoshikawa et al., 2004).

Our research site is located within a tundra ecosystem of dominant caribou lichen (*Cladonia mitis, Cladonia crispata, and Cladonia stellaris*), sphagnum moss (*Sphagnum magellanicum, Sphagnum angustifolium, and Sphagnum fuscum*) and tussock tundra (*Eriophorum vaginatum*) communities. Understory lichen, sphagnum moss, and tussock tundra occupied fractions of 27,





53, and 20 % of the site, respectively. Using a forced diffusion (FD) chamber system method
(Kim et al., 2016), soil $CO_2$ efflux was continuously measured at intact and crustose
lichen-infected sphagnum communities (64°51'42.8"N; 163°42'39.1"W; 42 a.s.l.m.) underlain
by discontinuous permafrost   in the tundra ecosystem of Council in western Alaska, during the
growing seasons of 2015 and 2016.
The annual average air temperature and precipitation were -3.2 °C and 394 mm, respectively, at
the Nome airport from 1907 to 2016. Air temperature ranged from 24.2 °C in June to -33.1 °C in
January in 2015, and from 22.4 °C in May to -27.1 °C in December in 2016 at Council. Annual
precipitation in 2015 and 2016 were 401 and 632 mm, respectively, including winter snowfall
(Western Regional Climate Center). During the growing season (June to September), average
ambient temperature and summed precipitation were $9.5 \pm 4.9$ °C and 272 mm in 2015 (Kim et
al., 2016), and $12.1 \pm 3.8$ °C and 597 mm in 2016, respectively. The precipitation in August to
September corresponds to 67 and 66 % of the entire growing seasons for 2015 and 2016,
respectively. Growing season temperature and precipitation for the past century was $8.4 \pm 2.5$ °C
and 215.4 mm, indicating cooler and much drier conditions. In other words, the growing periods
in 2015 and 2016 represent much hotter and wetter conditions than during the rest of the past
century. Our research site can be only be approached from early June to early October, as the
access road to Skookum Pass is kept closed during the snow-covered period by the Alaska
Department of Transportation.
Soil temperature was measured at 2 and 5-cm depths below the surface within intact and crustose
lichen-infected sphagnum moss colonies using two loggers with five sensors in 2015 and 2016
(logger: U12-006; sensor: TMC-HD, Onsetcomp, USA). Ambient temperature at 2.0 m above the
surface was also monitored at the site. Soil moisture at 2 and 5-cm depths below the surface of
each sphagnum colony was also measured with two loggers and four probes (logger: H21-002;
probe: SMD-M005, Onsetcomp, USA) in parallel with soil temperature (Figure 2), showing the
same period as observations of soil $CO_2$ efflux-measurement. Snow depth was monitored using
time-lapse camera at a six-hour interval from September 22, 2015 to June 13, 2016
(Supplementary Figure S2).



**2.2 Forced Diffusion (FD) CO$_2$ Efflux Chamber**
The FD CO$_2$ efflux chamber (Eosense, Canada) is a yearlong continuous soil CO$_2$
efflux-measuring system similar to a dynamic chamber, as described in Kim et al. (2016) in
detail. The FD structure consists of a single high-accuracy CO$_2$ sensor, an internal data-logger,
two valves, and a small diaphragm pump that operates only for short duration to bring a target air
sample to the sensor (Risk et al., 2011). The CO$_2$ sensor can determine a wide range efflux of 0
to 20 μmol m$^{-2}$ s$^{-1}$ at a measuring interval from 5 to 1440 min, under the ambient temperature of
-20 to 50 °C. The sensor is operated by a 12-volt power supply system including a cold-proofed
external battery (105-A AGM PVX-1040, USA), a 140-W solar panel (KD140GX-LFBS,
Kyocera Solar Inc., Japan), and a solar power charge converter (Morningstar S20 SunSaver,
USA). As shown in Supplementary Figure S1-a, we chose two target areas of intact and crustose
lichen-infected sphagnum communities, representing a relatively smooth and flat surface for
mounting the FD chamber on a previously installed soil collar (7.5-cm inside diameter; 9.0-cm
outside diameter; 10-cm height). The chamber was fixed with an attached mounting ring and four
legs. Two FD chambers were monitored from June 25, 2015 at the intact and crustose sphagnum
microsites. However, we could not determine the winter season CO$_2$ efflux during the
observation periods of 2015 and 2016 due to the heavy snow-covered solar panel by unexpected
winter storms. We confirmed heavy snowfall in early December of over 1.0 m using time-lapse
camera data.
As shown in Figure 1, we performed a test in sampling time between 10-min (with 30-min
average) and 30-min intervals at the intact sphagnum community from June 25 to July 23, 2015.
Soil CO$_2$ efflux at mean 30-min with 10-min intervals and at 30-min intervals was
$0.91 \pm 0.21$ μmol m$^2$ s$^{-1}$ and $0.90 \pm 0.20$ μmol m$^2$ s$^{-1}$, respectively, suggesting that there was no
significant difference, based on a one-way ANOVA at the 95 % confidence level ($p < 0.001$). As
a result, we set the 30-min sampling interval during the observation periods of 2015 and 2016, in
order to maintain low power consumption.



**2.3 Simulated Soil $CO_2$ Efflux**
We estimated the temperature sensitivity of soil $CO_2$ efflux collected by FD chamber by plotting
the exponential relationship between air temperature and soil temperature at depths of 2 and 5 cm,
in intact and crustose lichen-infected sphagnum moss colonies, by using the following equation:
$$CO_2 \ efflux = \beta_0 \times e^{\beta_1 \times T}, \qquad\qquad (1)$$
where $CO_2$ *efflux* is the *measured daily soil $CO_2$ efflux* ($\mu$mol m$^{-2}$ s$^{-1}$), $T$ is temperature (°C), and
$\beta_0$ and $\beta_1$ are constants. This exponential relationship is commonly used to represent soil carbon
flux as a function of temperature (Davidson et al., 1998; Xu and Qi, 2001; Davidson and
Janssens, 2006; Rayment and Jarvis, 2000; Kim et al., 2014a, 2014b, 2016). $Q_{10}$ temperature
coefficient values were calculated as in Davidson and Janssens (2006) and Kim et al. (2016):
$$Q_{10} = e^{\beta_1 \times 10}, \qquad\qquad (2)$$
$Q_{10}$ here is a measure of the change in reaction rate at intervals of 10 °C and is based on Van't
Hoff's empirical rule that a rate increase of 2 to 3 times occurs for every 10 °C rise in
temperature (Lloyd and Taylor, 1994).
A reference value for $R_{10} = \beta_0 \times e^{\beta_1 \times 10}$ (i.e., soil $CO_2$ efflux normalized to air temperature of
10 °C), where $\beta_0$ and $\beta_1$ are constants from equation (1), based on monthly calculations. Using
the calculated values for $Q_{10}$ and $R_{10}$, soil $CO_2$ efflux was simulated on the basis of the measured
air temperature. Simulated monthly soil $CO_2$ efflux values $R_i$ ($\mu$mol m$^{-2}$ s$^{-1}$) (as in Qi et al.
(2002); Curiel Yuste et al. (2004, 2005); Edwards et al. (2006); Gaumont-Guay et al. (2006,
2008); Begeron et al. (2007); Kim et al. (2014, 2016); and Makita (2017)), were calculated as:
$$R_i = R_{10} \times Q_{10}^{[(T-10)/10]}. \qquad\qquad (3)$$
The parameters of the nonrectangular hyperbola function were determined daily, using a
seven-day moving window and the least-squares method. Soil $CO_2$ efflux (SR) was estimated
using the following two models (Ueyama et al., 2014):



$$CO_2 \; efflux = R_0 \times Q_{10}^{(Ta/10)} ,\qquad\qquad (4)$$
$$CO_2 \; efflux = R_{ref} \times [\frac{E_0}{R_{gas}}\left(\frac{1}{T_k + T_{ref} - T_0} - \frac{1}{T_k + T_a - T_0}\right)], \qquad (5)$$
where $T_a$ is air temperature at 0.5 m, $R_o$ represents soil $CO_2$ efflux at 0 °C, and $Q_{10}$ is the
temperature sensitivity coefficient of soil $CO_2$ efflux. $R_{ref}$ is the soil $CO_2$ efflux at $T_{ref}$, $E_0$ is the
activation energy, and $R_{gas}$ is the ideal gas constant. $T_k$, $T_0$, and $T_{ref}$ are 273.15 K, 227.13 K, and
283.15 K, respectively (Lloyd and Taylor, 1994). We used the conventional $Q_{10}$ model to
estimate soil $CO_2$ efflux, but used the Lloyd and Taylor model equation (6) for uncertainty
estimates, as $Q_{10}$ exhibited clear seasonal variations, whereas $E_0$ showed no discernable seasonal
variation.
**3 Results and Discussion**
**3.1 Temporal Variations in Environmental Parameters**
Ambient air temperature at 2.0 m above the surface ranged from -33 °C to 24 °C for 2015, and
from -27 °C to 22 °C for 2016. Average air temperature was 10.7 and 11.6 °C during the growing
seasons (June to September) of 2015 and 2016, respectively, which was much higher than the
8.7 °C annual average air temperature during the growing seasons of 1960 and 2016. Figure 2
shows temporal variations in soil temperature and soil moisture at 2-cm (Figure 2a) and 5-cm
(Figure 2b) depths during the observation periods of 2015 and 2016. Soil temperature at 2-cm
depth was greater than at 5-cm depth during the growing seasons, indicating a significant
difference at intact sphagnum moss but no significant difference for the crustose sphagnum moss
community, based on a one-way ANOVA at the 95 % confidence level ($p < 0.05$). Soil
temperature at 2 cm for the intact sphagnum regime was higher than at the crustose colony,
representing a significant difference (95 % confidence level; $p < 0.05$); on the other hand, soil
temperature at 5 cm for intact sphagnum was lower than the crustose community, though not
significantly different (95 % confidence level; $p < 0.05$), as shown in Table 1.



From the strong linear relationship between air temperature and soil temperature, air temperature
accounts for 82 % and 76 % of variability in soil temperature at 2- and 5-cm depths during the
growing seasons of 2015 and 2016, respectively. Ambient temperature was a useful proxy for
soil temperature. The air temperature of 13.0 ± 1.9 °C in August of 2016 was much greater than
10.1 ± 2.7 °C in August of 2015, resulting in the significant difference in soil temperature at
2- and 5-cm depths in August between 2015 and 2016, based on a one-way ANOVA at the 95 %
confidence level ($p < 0.05$). This may have prompted the difference in soil $CO_2$ emission
between the Augusts of 2015 and 2016, as temperature is a key driver in regulating soil $CO_2$
production and emission to the atmosphere (Xu and Qi, 2001; Davidson and Janssens, 2006; Kim
et al., 2014b; 2016).
Peaks in soil moisture during the soil thawing of early May were found at 2- and 5-cm depths in
2015 and 2016 (Figure 2), suggesting the response from soil moisture at 2- and 5-cm depths for
intact sphagnum is much more sensitive to soil thawing than at crustose regime. This may reflect
the difference in moisture holding capacity between live and shriveled sphagnum. During the
observation periods of 2015 and 2016, soil moisture at 2- and 5-cm depths in intact sphagnum
moss cover was explicitly higher than in crustose sphagnum moss patch, indicating a significant
difference, based on a one-way ANOVA at the 95 % confidence level ($p < 0.05$). Soil moisture at
2-cm depth was lower than 5-cm depth at the intact sphagnum moss colony, showing a
significant difference based on a one-way ANOVA at the 95 % confidence level ($p < 0.05$).
However, at the crutose lichen-infected sphagnum moss regime, soil moisture at 2-cm depth was
similar to those at 5-cm depth, representing no significant difference based on a one-way
ANOVA at the 95 % confidence level ($p < 0.05$). This reflects the lower, analogous soil moisture
between 2- and 5-cm depths of crustose moss relative to intact sphagnum moss, proving that the
forfeiture of essentially physiological and ecological functions occurs by the airborne infection of
crustose lichen (*O. frigida*) on healthy sphagnum moss.
Soil moisture was sensitive to rainfall events during the growing seasons of 2015 and 2016. Soil
thawing timing can detect a sudden rise of soil moisture at 2- and 5-cm depths in intact and
crustose sphagnum moss communities in early spring of 2015 and 2016 (Figure 2b), in parallel to



a sharp jump of soil temperature at 2-cm depth at both sphagnum moss regimes (Figure 2a). We
computed thawing rates between 2- and 5-cm depths when soil moisture was over 0.20 m$^3$ m$^{-3}$,
representing thawing rates in the early spring of 2015 and 2016 of 0.75 and 0.27 cm day$^{-1}$ at
intact sphagnum moss. On the other hand, thawing rates at crustose sphagnum moss between the
two depths are nearly 0 cm day$^{-1}$. This demonstrates the crustose lichen-infected sphagnum moss
loses the soil moisture holding capacity by causing the withering and death of intact sphagnum
moss. However, the mean thawing rate of 0.438 cm day$^{-1}$ is comparable with those in this study
during the growing seasons of 2011 to 2014 obtained at neighboring sites (Kim et al., 2016).
When soil temperature drops to below zero during the late growing season of 2015, soil moisture
falls sharply at 2-cm depth in intact (0.24 to 0.16 m$^3$ m$^{-3}$) and crustose (0.22 to 0.05 m$^3$ m$^{-3}$;
Figure 2a), respectively. Ironically, soil moisture at 2-cm depth in crustose sphagnum moss has
maintained higher levels than in intact sphagnum moss since August of 2016; on the other hand,
soil moisture at 5-cm depth in crustose sphagnum moss since the late growing season of 2016 is
lower than the intact sphagnum moss community. The latter demonstrates natural phenomena as
shown in 2015 (Figure 2a and 2b).
These changes in daily snow accumulation and ablation are documented by time-lapse camera at
six-hour intervals from September 22, 2015 to June 13, 2016, as shown in Supplemental Figure
S2. Snow-covered day and snow-disappearance day are November 3, 2015 and May 6, 2016,
respectively, based on the criteria that 1) lingering snowpack cover exceeds fifteen consecutive
days upon the snow-covered day, and 2) less than half of the surface is covered by snowpack
according to the naked eye upon the snow-disappearance day.
**3.2 Seasonal Variations in Soil CO$_2$ Emissions**
Soil CO$_2$ efflux-measurement was initiated on intact and crustose lichen-infected sphagnum moss
communities beginning June 25, 2015. CO$_2$ emissions and air temperature were measured with
the FD chamber system at both sphagnum moss regimes from June 25 to November 9, 2015, and
from June 18 to September 28, 2016, respectively (Figure 3). Unfortunately, we could not
determine the winter season soil CO$_2$ emission, due to shutoff of the solar panel power supply by
unexpected deeper snowfall in the early winter of 2015. Average growing season soil CO$_2$



effluxes at intact and crustose sphagnum moss regimes were $0.39 \pm 0.18$ and
$0.38 \pm 0.22$ µmol m$^{-2}$ s$^{-1}$ for 2015, and $0.38 \pm 0.21$ and $0.42 \pm 0.27$ µmol m$^{-2}$ s$^{-1}$ for 2016,
respectively (Table 1). The difference in soil $CO_2$ effluxes between intact and crustose for the
first efflux-measuring year (2015) was not significant, based on a one-way ANOVA at the 95 %
confidence level ($p < 0.05$). However, the difference between the regimes for the second year
(2016) was significant (95 % confidence level, $p < 0.05$), indicating the average ratio of crustose
to intact soil $CO_2$ effluxes was $1.70 \pm 1.27$ (Table 1), a distinct increment of 70 % compared to
intact soil $CO_2$ emissions during the growing season of 2016. Responses from soil $CO_2$ efflux at
intact sphagnum moss to crustose sphagnum moss showed positively linear relationships (*Intact*
*$CO_2$ = 0.98 × Crustose $CO_2$ – 0.01; $R^2$ = 0.73 for 2015; Intact $CO_2$ = 0.51 × Crustose*
*$CO_2$ + 0.24; $R^2$ = 0.17 for 2016)*, as shown in Figure 4. This implies that higher soil $CO_2$ efflux
during the growing season of 2016 is associated with enhanced decomposition of organic matter
at crustose lichen-infected sphagnum moss under a hotter and drier soil environment (Figure 2),
relative to the intact sphagnum moss community.
There is little data on $CO_2$ efflux-measurements from crustose lichen-infected sphagnum moss,
which may indicate a lack of attention toward the ecological and climate impacts upon crustose
lichen in Arctic terrestrial ecosystems. On the other hand, biological soil crusts (BSCs), which
are the first organisms to colonize the exposed soil surface, inhabit an organic layer less than 0.01
m thick in the early stage of primary succession after glaciers retreated (Belnap and Lange, 2003).
Also, BSCs consist of the organic residues from lichen, moss, and cyanobacteria through the
successional stages after deglaciation. Nakatsubo et al. (1998), Yoshitake et al. (2007), and Chae
et al. (2016) measured soil microbial respiration on the BSCs with black color (BSCs-B),
including soil surface communities consisting of blackish organic residues in Ny-Ålesund,
Svalbard, Norway, ranging from 0.21 to 0.35 µmol m$^{-2}$ s$^{-1}$. These values are similar to the results
obtained in this study; however, previous results were determined by the manual chambers when
they infrequently visited the sites in summer (Savage and Davidson, 2005). Depending on the
observation schedule and field sites, the low-data approach can suffice for seasonal totals, but
may lead to critical episodic and process-driven events being missed or misinterpreted. Parkin
and Kaspar (2003) offered a detailed study on the effect of measuring frequency, demonstrating



that using a scheduled daily measurement for $CO_2$ efflux-estimates can result in a deviation of up
to 30 % from the daily average. The net impact this bias has on estimated effluxes depends on the
daily emission range, meaning that the estimation error will change with environmental and
seasonal trends (Savage and Davidson, 2005; Kim et al., 2016).

**3.3 Sensitivity of Soil $CO_2$ Emissions to Temperature and Soil Moisture**

Responses in soil $CO_2$ efflux observed at intact and crustose lichen-infected sphagnum moss
communities to temperature in air and soil at 2- and 5-cm depths during the observation periods
of 2015 and 2016 are shown in Figure 5. Soil $CO_2$ efflux follows the normal exponential
relationship to temperature as in the equation (1). In terms of month-based $Q_{10}$ value as listed in
Table 2, the values in June and July of 2015 are much lower than other months, due to nearly
fixed soil $CO_2$ effluxes at intact and crustose sphagnum moss communities relative to changes in
temperature (Figure 3). Furthermore, the highest $Q_{10}$ values in September of 2016 at intact
sphagnum moss are 10.8, 17.3, and 48.6 for the temperature in air and soil 2- and 5-cm depths,
respectively. The greatest $Q_{10}$ values in August of 2016 at crustose sphagnum moss were 3.32,
15.9, and 16.3 for the temperature in air and soil 2- and 5-cm depths, respectively. This suggests
a seasonal dependence of soil $CO_2$ efflux on temperature for two sphagnum moss patches.
Average temperature in air and soil at 2- and 5-cm depths elucidates over 60 % of variability in
soil $CO_2$ effluxes at intact and crustose sphagnum moss for 2015; however, the sensitivity of soil
$CO_2$ effluxes to temperature for 2016 was much lower than for 2015.
During the observation periods of 2015 and 2016, soil temperature at 2- and 5-cm depths is
strongly dependent on seasonal variations in air temperature. Temperature is a most significant
driver in modulating soil $CO_2$ emission in terrestrial ecosystems (Davidson et al., 1998; Xu and
Qi, 2001; Davidson and Janssens, 2006; Rayment and Jarvis, 2000; Bond-Lamberty and
Thomson, 2010; Kim et al., 2014a, 2014b, 2016). On the other hand, soil moisture is an
important parameter in constraining $CO_2$ emissions in the intact sphagnum moss community,
tundra ecosystems of west Alaska during the growing seasons of 2011 and 2012 (Kim et al.,
2014b), and other terrestrial ecosystems (Davidson and Janssens, 2006; Davidson et al., 1998;
Oberbauer et al., 2007; Jansen et al., 2014). Although there was heavy rain for August and



September of 2016 (393.5 mm) compared to 2015 (181.5 mm), observed soil $CO_2$ effluxes at
intact and crustose sphagnum moss communities were not lower than 2015. This may be due to a
loss of water-retaining capacity at the crustose lichen-infected sphagnum moss regime, with
higher soil $CO_2$ effluxes than the intact sphagnum moss in the latter half of the 2016 growing
season. Moreover, we found that soil moisture content at 5-cm depth in intact sphagnum moss is
much greater than in the crustose sphagnum moss colony since August of 2016, as shown in
Figure 2b. Therefore, while soil moisture acts as a well-known key role in restraining soil $CO_2$
emissions in the intact sphagnum moss community, soil moisture in crustose sphagnum moss is
not prompted to emit soil carbon to the atmosphere (Table 1).
The correlation coefficients ($R^2$) for temperature in air and soil at intact and crustose sphagnum
moss of 2015 were higher than 2016. We found distinct difference in the response from soil $CO_2$
efflux to air and soil temperature at 2- and 5-cm depths. $Q_{10}$ values at intact and crustose
sphagnum moss during 2015 and 2016 can be estimated by equation (2). $Q_{10}$ value increases with
soil depth, indicating that the extent of soil temperature at deeper soil depth appears much
narrower than at shallower depth (Mikan et al., 2004; Pavelka et al., 2007; Kim et al., 2014; Kim
et al., 2016).
During the growing seasons of 2015 and 2016, Figure 6 shows seven-day moving $Q_{10}$ values,
calculated for each of research plots using equation (6). Using average two-growing-season $Q_{10}$
values ± standard deviation for air temperature, soil temperature at 2- and 5-cm depths are
$2.37 \pm 0.68$, $2.25 \pm 0.72$, and $2.16 \pm 0.58$ at intact, and $2.31 \pm 0.65$, $2.44 \pm 0.63$, and $2.59 \pm 0.77$
at crustose sphagnum moss, respectively. Furthermore, $Q_{10}$ values at crustose sphagnum moss are
wildly more fluctuant than $Q_{10}$ values at intact sphagnum moss in late growing seasons of 2015
and 2016.
**3.4 Estimation of Simulated Soil $CO_2$ Efflux**
Based on $Q_{10}$ and $R_{10}$ relationships (equation 2 ~~and 3~~), simulated daily soil $CO_2$ effluxes at intact
and crustose sphagnum moss communities were estimated using equation (3) and *in-situ*
temperature in air and soil at 2- and 5-cm depths from June 25, 2015 to September 28, 2016, with



temporal variation in air temperature (Figure 7). Temporal variations in simulated soil $CO_2$
effluxes are synchronized with seasonal variation of ambient temperature, reflecting that soil
temperature at 2- and 5-cm depths accounted for 92 and 82 % of the variability in air temperature
at intact sphagnum, and 88 and 82 % of the variability in air temperature at the crustose
sphagnum moss colony for 2015, respectively. For 2016, soil temperature at 2- and 5-cm depths
elucidated 90 and 76 % of the variability in air temperature at intact sphagnum, and 81 and 80 %
of the variability in air temperature at crustose sphagnum moss community, respectively. Air
temperature is an important key in stimulating soil temperature in terrestrial ecosystems (Kim et
al., 2014a, 2016). The relationships between observed and simulated daily soil $CO_2$ effluxes were
positively linear during the two growing seasons of 2015 and 2016, as shown in Figure 8. This
suggests that the observed soil $CO_2$ effluxes account for 64, 70, and 72 % of the variability in
daily soil $CO_2$ effluxes simulated by temperature in air and soil at 2- and 5-cm depths at intact
sphagnum moss cover, and the observed soil $CO_2$ effluxes explain 48, 63, and 60 % of the
variability in simulated daily soil $CO_2$ effluxes by three temperatures at the crustose sphagnum
moss colony, respectively. During the two growing seasons of 2015 and 2016, the difference
between observed and simulated soil $CO_2$ effluxes by temperature in air and soil 2- and 5-cm
depths at two sphagnum moss colonies were significantly different, based on a one-way ANOVA
at the 95 % confidence level ($p < 0.05$). However, the difference between observed and
simulated soil $CO_2$ effluxes from air temperature at intact sphagnum moss cover for 2016 is not
explicitly significant difference ($p < 0.05$).
Average simulated monthly soil $CO_2$ efflux was also computed and is listed in Table 3, showing
the seasonal pattern and including the low rate of $CO_2$ emission that can be expected overwinter
during snow-covered period (186 days), as described in section 3.1 and shown in Supplemental
Figure S2. Although non-growing season soil $CO_2$ efflux by FD chamber was not measured in
this study, determining the annual carbon budget using simulated daily soil $CO_2$ efflux for three
temperatures with time-lapse camera data, we can establish seasonal budgets during
snow-covered and snow-free periods. Simulated soil $CO_2$ effluxes in intact sphagnum moss are
13.7, 22.0, and 22.5 gC m$^{-2}$ period$^{-1}$ for temperature in air and soil at 2- and 5-cm depths during
the snow-covered period, corresponding to 20.0, 30.5, and 34.8 % of annual simulated carbon





emissions, respectively. The winter-simulated soil $CO_2$ effluxes in crustose lichen-infected
sphagnum moss are 10.4, 16.8, and 17.1 gC m$^{-2}$ period$^{-1}$ for three temperatures, corresponding to
16.2, 28.4, and 30.4 % of annual simulated carbon emission, respectively. On the other hand,
during the snow-free period, average simulated soil $CO_2$ effluxes in intact sphagnum moss are
57.1, 50.2, and 41.9 gC m$^{-2}$ period$^{-1}$ for temperature in air and soil 2- and 5-cm depths,
corresponding to 80.0, 69.5, and 65.2 % of annual carbon emissions, respectively. Further, the
simulated soil $CO_2$ effluxes in crustose lichen-infected sphagnum moss are 55.7, 43.8, and
40.5 gC m$^{-2}$ period$^{-1}$ for three temperatures, corresponding to 83.8, 71.6, and 69.6 % of annual
simulated carbon emission, respectively.
On the whole, 28.4 % and 25.0 % of annual simulated soil $CO_2$ effluxes in intact and crustose
sphagnum moss patches (respectively) were likely emitted through the snowpack to the
atmosphere during the non-growing season with the remainder during the growing season. Many
previous studies on winter soil $CO_2$ efflux-measurement have represented similar aspects, and
winter contributions to soil $CO_2$ emission have generally elucidated 10 to 30 % of annual carbon
budgets for tundra (Oechel et al., 1997; Fahnestock et al., 1998; Björkman et al., 2010; Kim et al.,
2013; 2016), alpine and subalpine forests (Brooks et al., 1996; Mast et al., 1998; Monson et al.,
2006), and boreal forests (Winston et al., 1997; Kim et al., 2007; 2013; Kim, 2014).
Kim et al. (2014b) found the deviation between the manual chamber and continuous
measurement by FD chamber methods as high as 47 %. This may be due to differences in
measuring method and frequency under sunny sky (manual) compared to year-long and
continuous (FD). The additional measuring frequency possible with FD could cause some
re-evaluation of interpreted annual carbon budgets at representative spots, and would aid in
applying terrestrial ecosystem models (e.g., land surface models (LSMs)) to high time-resolution
data. Therefore, continuous monitoring of soil $CO_2$ efflux-measurement using FD chambers
initiates new fields of opportunity and understandings. As drastic climate warming enhances
permafrost degradation in the Subarctic and Arctic, we will reckon with large stocks of ancient
soil carbon that will become available for microbial activation (Schuur et al., 2009; Tarnocai et
al., 2009; Grosse et al., 2011), as well as other ecological and biogeochemical significance



(Walter et al., 2008; Schuur et al., 2009; Zona et al., 2009; Sachs et al., 2010; Lawrence et al.,
2015; Natali et al., 2015) across the landscape. Therefore, yearlong soil $CO_2$ efflux using FD
chamber systems will be required to pursue concurrent changes in carbon storage response to a
microbial outbreak in the Arctic-wide distributed extents of the sphagnum moss regime (Whalen
and Reeburgh, 1998).

## 4 Conclusions

Soil $CO_2$ efflux measurement is an important component for estimating annual carbon budgets in
response to changes in increasing ambient temperature, thawing permafrost, and snow-covered
extent in the Subarctic and Arctic. Here, continuous monitoring of soil $CO_2$ efflux using a forced
diffusion (FD) chamber system was performed at intact and crustose lichen (*Ochrolecia*
*frigida*)-infected sphagnum moss communities of tundra ecosystem in western Alaska during the
growing seasons of 2015 and 2016. Temperature was a key driver in governing soil $CO_2$ efflux at
two sphagnum moss patches during the observation periods of 2015 and 2016. Furthermore,
ambient temperature elucidates over 80 % of the variability in soil temperature at 2- and 5-cm
depths during those two growing seasons. At the crustose sphagnum moss community, the
differences in soil temperature and soil moisture at 2- and 5-cm depths are not explicit,
suggesting the loss of ecological and thermal functions. Thus, soil moisture plays a significant
role in retraining soil $CO_2$ emission in healthy sphagnum moss carpets (Davidson et al., 1998;
Davidson and Janssens, 2006; Oberbauer et al., 2007; Jansen et al., 2014; Kim et al., 2014b).
Further, soil moisture in withered sphagnum moss patches is not so much as a limiter as a
stimulator for soil carbon emission. Responses from soil $CO_2$ efflux at intact sphagnum moss to
crustose sphagnum moss patches show positive linear relationships, indicating that soil $CO_2$
efflux at crustose sphagnum moss explains 73 % and 17 % of variability in soil $CO_2$ efflux at the
intact sphagnum moss colony for 2015 and 2016, respectively. This implies that high soil $CO_2$
efflux during the growing season of 2016 resulted from enhanced decay of soil organic matter at
crustose lichen-infected sphagnum moss under the hot and moist soil environment relative to the
intact sphagnum moss community. This finding thus demonstrates the shriveled sphagnum moss
colony is an atmospheric $CO_2$ source reservoir, and that the degradation of permafrost will be




stimulated by the widespread outbreak of airborne crustose lichen on the healthy sphagnum moss
community response to rapid climate change in the Subarctic and Arctic.
Simulated daily soil $CO_2$ effluxes at intact and crustose lichen-infected sphagnum moss
communities were estimated using *in-situ* temperature in air and soil at 2- and 5-cm depths from
June 25, 2015 to September 28, 2016, with temporal variation in air temperature, which can
discriminate between seasonally snow-covered and snow-free soil carbon emissions. Time-lapse
camera data provides us beneficial information for the snow-covered period of 185 days and the
snow-free period. Average winter soil $CO_2$ effluxes at intact and crustose sphagnum moss
communities are 19.4 and 15.3 gC m$^{-2}$ period$^{-1}$, respectively, corresponding to 28.4 and 20.0 % of
annual simulated carbon emission, with the remainder during the snow-free period. These values
are equivalent to 10 to 30 % of the annual carbon budget observed in various tundra ecosystems.
At the crustose lichen-infected sphagnum moss colony, daily soil $CO_2$ effluxes simulated by
temperature will be underestimated due to lack of consideration of additional contributions from
soil $CO_2$ efflux regardless of the effect of soil moisture. However, at the intact sphagnum moss
regime, simulated daily soil $CO_2$ effluxes will be relatively overestimated owing to no regard of
constrained soil $CO_2$ efflux by the influence of soil moisture. However, as conducted by Risk et
al. (2011), the monitoring of soil $CO_2$ efflux must also show representative points during the
snow-covered and snow-free periods, along with the monitoring of environmental parameters
within the sites.
In conclusion, these findings imply that soil $CO_2$ emission at a crustose lichen-infected sphagnum
moss community will be gradually enhanced by the wide spread of aerial plants on flawless
sphagnum moss patches, the subsequently increased decay of soil organic matter, and the rapid
degradation of permafrost, in response to recent and drastic changes in climate and environment
in the Subarctic and Arctic.



**Acknowledgments**
This research was supported by a National Research Foundation of Korea Grant from the Korean
Government (MSIT; the Ministry of Science and ICT) (NRF-2016M1A5A1901769)
(KOPRI-PN19081) (Title: Circum Arctic Permafrost Environment Change Monitoring, Future
Prediction and development Techniques of useful biomaterials (CAPEC Project)). Mr. N. Bauer
of the International Arctic Research Center (IARC) at the University of Alaska Fairbanks
provided constructive editorial revisions for the manuscript.





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



**Figure Legends**
**Figure 1**. The linear relationship of soil $CO_2$ efflux measured by forced diffused (FD) chamber
system between each ten-min (with thirty-min average) and thirty-min interval at intact
sphagnum community from June 25 to July 23, 2015. This suggests there is no significant
difference, based on a one-way ANOVA at the 95 % confidence level ($p < 0.001$). The dashed
line denotes a 1:1 line.
**Figure 2**. Temporal variations in soil temperature (solid line) and soil moisture (dotted) at a)
2-cm depth and b) 5-cm depth at intact and crustose sphagnum moss regimes during the
observation periods of 2015 and 2016.
**Figure 3**. Temporal variations in mean daily soil $CO_2$ effluxes with standard deviation (95 %
confidence level) and ambient temperature at intact and crustose sphagnum moss colonies during
the observation periods of 2015 and 2016.
**Figure 4**. Responses from soil $CO_2$ effluxes at intact to crustose sphagnum moss during a) 2015
(circles) and b) 2016 (squares). The thin dotted line indicates a 1:1 line. Correlation curves for
2015 and 2016 are shown by solid and dotted lines, respectively.
**Figure 5**. Responses from mean daily soil $CO_2$ effluxes to air temperature (pluses), soil
temperature at 2 cm (triangles), and 5 cm (grey circles) below the surface at a) intact and b)
crustose for 2015, and c) intact and d) crustose for 2016. Correlation curves for air temperature
and soil temperature at 2- and 5-cm depths are shown by solid, dashed, and dotted lines,
respectively.
**Figure 6**. Temporal variations in $Q_{10}$ values using equation (6) for air temperature (solid line),
soil temperature at 2 cm (dotted), and 5 cm (grey) below the surface at a) intact and b) crustose
for 2015, and c) intact and d) crustose for 2016. $Q_{10}$ values observed at crustose sphagnum moss
for September of 2015 and 2016 show much wider fluctuation than at intact sphagnum moss.
**Figure 7**. Temporal variations in soil $CO_2$ efflux simulated by equation (3) and air temperature at





a) intact and b) crustose sphagnum moss from June 25, 2015 to September 30, 2016. Shaded
columns represent the observation periods of 2015 and 2016.
**Figure 8**. Relationship between mean daily observed soil $CO_2$ effluxes ($OSR_{intact}$ and $OSR_{crustose}$)
and simulated soil $CO_2$ efflux (SSR) based on air temperature (AT), soil temperature at 2- (ST2)
and 5-cm (ST5) depths at 1) intact and 2) crustose sphagnum moss. Thin dotted lines indicate a
1:1   line.   In   Figure   a),   $OSR_{intact} = 1.02\ SSR_{AT} + 0.05$   ($R^2 = 0.64$)   (solid   line),
$OSR_{intact} = 1.34\ SSR_{ST2} + 0.01$   ($R^2 = 0.70$)   (dashed   line),   and   $OSR_{intact} = 2.24\ SSR_{ST5} - 0.12$
($R^2 = 0.72$) (dotted line), and in Figure b), $OSR_{crustose} = 0.95\ SSR_{AT} + 0.06$ ($R^2 = 0.48$) (solid line),
$OSR_{crustose} = 1.48\ SSR_{ST2} + 0.01$   ($R^2 = 0.63$)   (dashed   line),   and   $OSR_{crustose} = 1.74\ SSR_{ST5} - 0.04$
($R^2 = 0.60$) (dotted line), respectively.




Table 1. Monthly mean (standard deviation) in $CO_2$ efflux, ratio of crustose to intact efflux (C/I), and soil temperature and soil moisture at 2 and 5 cm depths in intact and crustose sphagnum moss communities during the growing seasons of 2015 and 2016

| Year | Month | $CO_2$ efflux ($\mu$mol m$^{-2}$ s$^{-1}$) | | C/I Ratio | Soil temperature (°C) | | | | Soil moisture (m$^3$ m$^{-3}$) | | | |
| | | Intact | Crustose | | Intact | | Crustose | | Intact | | Crustose | |
| | | | | | 2 cm | 5 cm | 2 cm | 5 cm | 2 cm | 5 cm | 2 cm | 5 cm |
| 2015 | June* | 0.45 (0.09) | 0.42 (0.11) | 0.93 | 10.5 (2.13) | 7.53 (1.18) | 9.94 (1.51) | 8.93 (1.18) | 0.22 (0.11) | 0.21 (0.02) | 0.20 (0.02) | 0.24 (0.03) |
| | July | 0.53 (01.0) | 0.51 (0.15) | 0.97 | 13.0 (2.21) | 9.59 (1.62) | 12.4 (1.93) | 11.3 (1.71) | 0.23 (0.01) | 0.23 (0.02) | 0.21 (0.01) | 0.23 (0.02) |
| | August | 0.42 (0.16) | 0.41 (0.22) | 0.95 | 9.27 (1.50) | 7.24 (0.98) | 8.79 (1.33) | 8.26 (1.04) | 0.25 (0.02) | 0.24 (0.03) | 0.23 (0.02) | 0.25 (0.04) |
| | September | 0.21 (0.13) | 0.19 (0.16) | 0.86 | 2.68 (3.32) | 2.24 (2.39) | 2.28 (3.29) | 2.35 (2.83) | 0.26 (0.03) | 0.28 (0.04) | 0.22 (0.05) | 0.25 (0.04) |
| | Growing season # | 0.39 (0.18) | 0.38 (0.22) | 0.93 | 8.49 (4.94) | 6.47 (3.54) | 8.02 (4.86) | 7.45 (4.24) | 0.24 (0.03) | 0.24 (0.03) | 0.22 (0.03) | 0.24 (0.03) |
| 2016 | June** | 0.27 (0.07) | 0.47 (0.22) | 2.01 | 12.5 (1.72) | 9.19 (1.24) | 11.9 (1.14) | 10.6 (1.09) | 0.23 (0.01) | 0.20 (0.01) | 0.23 (0.02) | 0.21 (0.03) |
| | July | 0.45 (0.17) | 0.52 (0.21) | 1.36 | 12.8 (1.88) | 10.1 (1.34) | 12.1 (1.65) | 11.3 (1.37) | 0.22 (0.03) | 0.21 (0.04) | 0.22 (0.03) | 0.20 (0.02) |
| | August | 0.50 (0.22) | 0.51 (0.33) | 1.13 | 11.3 (1.52) | 8.91 (1.11) | 11.0 (2.09) | 10.3 (1.73) | 0.26 (0.02) | 0.30 (0.02) | 0.29 (0.04) | 0.25 (0.05) |
| | September** | 0.21 (0.15) | 0.23 (0.15) | 1.99 | 4.34 (2.79) | 3.73 (2.05) | 3.93 (2.78) | 3.98 (2.41) | 0.26 (0.01) | 0.31 (0.04) | 0.29 (0.03) | 0.25 (0.03) |
| | Growing season # | 0.38 (0.21) | 0.43 (0.27) | 1.70 | 10.0 (4.06) | 7.87 (2.98) | 9.53 (4.04) | 8.91 (3.52) | 0.24 (0.03) | 0.26 (0.06) | 0.26 (0.04) | 0.23 (0.03) |

* The peridof 2015 is June 25 to 30.
** The period of 2016 is June 18 to 30 and September 1 to 28.
# The growing season denotes June to September of 2015 and 2016.



Table 2. $Q_{10}$ values and correlaton coefficients in the exponential equation for soil $CO_2$ efflux response to temperature in intact and crustose sphagnum moss communities of tundra , western Alaska during the observeration periods of 2015 and 2016, for which is the the equation is $CO_2$ efflux = $\beta_0$ x $\exp^{(\beta_1 \times T)}$, based on a one-way ANOVA at the 95% confidence level

| Year | Month | Depth | Intact | | Crutose | |
|---|---|---|---|---|---|---|
| Year | | (cm) | $Q_{10}$ | $R^2$ | $Q_{10}$ | $R^2$ |
| 2015* | June+July | Air 50 | 1.15 | 0.05 | 0.90 | 0.01 |
| | | 2 | 1.25 | 0.07 | 1.34 | 0.03 |
| | | 5 | 1.44 | 0.10 | 1.28 | 0.02 |
| | August | Air 50 | 3.51 | 0.53 | 6.34 | 0.37 |
| | | 2 | 3.53 | 0.49 | 7.99 | 0.47 |
| | | 5 | 5.89 | 0.47 | 9.38 | 0.38 |
| | September | Air 50 | 2.18 | 0.50 | 2.29 | 0.23 |
| | | 2 | 3.90 | 0.44 | 4.80 | 0.30 |
| | | 5 | 6.12 | 0.41 | 5.46 | 0.26 |
| | Oct + Nov | Air 50 | 1.18 | 0.01 | 3.48 | 0.38 |
| | | 2 | 1.47 | 0.03 | 6.01 | 0.44 |
| | | 5 | 1.43 | 0.01 | 11.40 | 0.33 |
| | Mean | Air 50 | 2.42 | 0.61 | 3.10 | 0.59 |
| | | 2 | 2.82 | 0.65 | 3.87 | 0.64 |
| | | 5 | 4.29 | 0.65 | 4.53 | 0.60 |
| 2016** | June+July | Air 50 | 1.27 | 0.02 | 0.83 | 0.11 |
| | | 2 | 2.00 | 0.08 | 2.01 | 0.03 |
| | | 5 | 3.79 | 0.17 | 1.42 | 0.01 |
| | August | Air 50 | 3.16 | 0.12 | 3.32 | 0.07 |
| | | 2 | 5.92 | 0.17 | 15.90 | 0.42 |
| | | 5 | 5.34 | 0.08 | 16.30 | 0.30 |
| | September | Air 50 | 10.80 | 0.56 | 1.55 | 0.05 |
| | | 2 | 17.30 | 0.47 | 2.23 | 0.09 |
| | | 5 | 48.60 | 0.47 | 2.03 | 0.05 |
| | Mean | Air 50 | 3.88 | 0.45 | 2.05 | 0.16 |
| | | 2 | 4.46 | 0.45 | 3.30 | 0.43 |
| | | 5 | 7.88 | 0.46 | 3.59 | 0.40 |

* The measuring period of 2015 is from June 25 to November 9.

** The  period of 2016 is from June 18 to September 28.





Table 3. Observed and simulated $CO_2$ efflux based on temperature in intact and crustose sphagnum moss communities during 2015 and 2016

| Date | $CO_2$ efflux ($\mu$mol m$^{-2}$ s$^{-1}$) | | | | | | | |
|---|---|---|---|---|---|---|---|---|
| | Observed | Simulated | | | Observed | Simulated | | |
| (mm-yy) | Intact | Air | 2 cm | 5 cm | Crustose | Air | 2 cm | 5 cm |
| Jul-15 | 0.53 (0.10) | 0.46 (0.11) | 0.40 (0.07) | 0.29 (0.04) | 0.51 (0.15) | 0.47 (0.13) | 0.37 (0.13) | 0.33 (0.05) |
| Aug-15 | 0.42 (0.16) | 0.31 (0.07) | 0.29 (0.06) | 0.24 (0.04) | 0.41 (0.22) | 0.30 (0.08) | 0.26 (0.06) | 0.24 (0.05) |
| Sep-15 | 0.21 (0.13) | 0.18 (0.08) | 0.16 (0.06) | 0.15 (0.04) | 0.19 (0.16) | 0.16 (0.08) | 0.14 (0.05) | 0.14 (0.05) |
| Oct-15 | 0.14 (0.08) | 0.13 (0.05) | 0.12 (0.04) | 0.13 (0.02) | 0.14 (0.12) | 0.11 (0.05) | 0.10 (0.03) | 0.10 (0.02) |
| Nov-15 | 0.17 (0.06) | 0.08 (0.04) | 0.12 (0.01) | 0.12 (0.01) | 0.09 (0.04) | 0.06 (0.04) | 0.09 (0.01) | 0.09 (0.01) |
| Dec-15 | N.D. | 0.04 (0.03) | 0.11 (0.01) | 0.11 (0.01) | N.D. | 0.03 (0.02) | 0.09 (0.01) | 0.09 (0.01) |
| 2015** | 0.39 (0.19) | 0.32 (0.15) | 0.28 (0.11) | 0.23 (0.07) | 0.37 (0.20) | 0.31 (0.16) | 0.26 (0.11) | 0.24 (0.09) |
| Jan-16 | N.D.# | 0.06 (0.03) | 0.11 (0.01) | 0.12 (0.01) | N.D. | 0.05 (0.02) | 0.09 (0.01) | 0.09 (0.01) |
| Feb-16 | N.D. | 0.06 (0.01) | 0.11 (0.01) | 0.12 (0.01) | N.D. | 0.05 (0.02) | 0.09 (0.01) | 0.09 (0.01) |
| Mar-16 | N.D. | 0.04 (0.02) | 0.11 (0.01) | 0.11 (0.01) | N.D. | 0.03 (0.02) | 0.08 (0.01) | 0.08 (0.01) |
| Apr-16 | N.D. | 0.12 (0.03) | 0.12 (0.01) | 0.12 (0.01) | N.D. | 0.10 (003) | 0.10 (0.01) | 0.10 (0.01) |
| May-16 | N.D. | 0.29 (0.19) | 0.23 (0.09) | 0.16 (0.05) | N.D. | 0.28 (0.22) | 0.18 (0.08) | 0.15 (0.06) |
| Jun-16 | 0.27 (0.07)* | 0.40 (0.12) | 0.34 (0.07) | 0.25 (0.04) | 0.47 (0.22)* | 0.40 (0.14) | 0.29 (0.08) | 0.26 (0.06) |
| Jul-16 | 0.45 (0.17) | 0.45 (0.12) | 0.39 (0.06) | 0.30 (0.04) | 0.52 (0.21) | 0.46 (0.14) | 0.36 (0.06) | 0.33 (0.04) |
| Aug-16 | 0.50 (0.22) | 0.40 (0.07) | 0.34 (0.05) | 0.27 (0.03) | 0.51 (0.33) | 0.39 (0.08) | 0.32 (0.07) | 0.30 (0.05) |
| Sep-16 | 0.21 (0.15) | 0.22 (0.08) | 0.19 (0.05) | 0.17 (0.03) | 0.24 (0.15) | 0.20 (0.09) | 0.16 (0.05) | 0.16 (0.04) |
| 2016** | 0.40 (0.22) | 0.36 (0.13) | 0.31 (0.10) | 0.25 (0.06) | 0.43 (0.28) | 0.36 (0.15) | 0.28 (0.10) | 0.27 (0.09) |

* The observed value is June 18 to 30, 2015.

** denote growing season (July to September) of 2015 and 2016.

# indicates not determined.





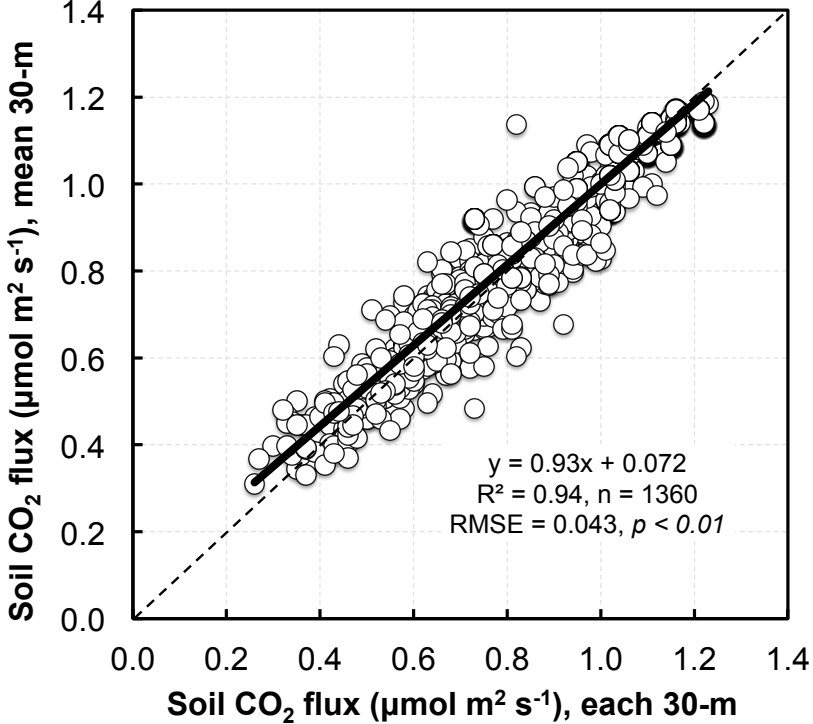

**Figure 1**. The linear relationship of soil $CO_2$ efflux measured by forced diffused (FD) chamber
system between each ten-min (with thirty-min average) and thirty-min interval at intact
sphagnum community from June 25 to July 23, 2015. This suggests there is no significant
difference, based on a one-way ANOVA at the 95 % confidence level ($p < 0.001$). The dashed
line denotes a 1:1 line.





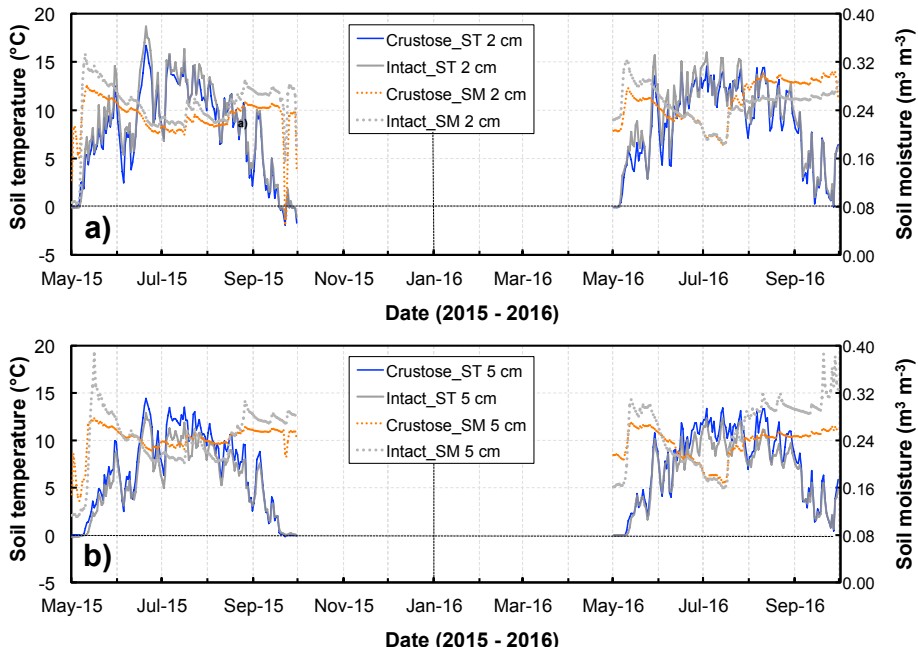

**Figure 2**. Temporal variations in soil temperature (solid line) and soil moisture (dotted) at a)
2-cm depth and b) 5-cm depth at intact and crustose sphagnum moss regimes during the
observation periods of 2015 and 2016.





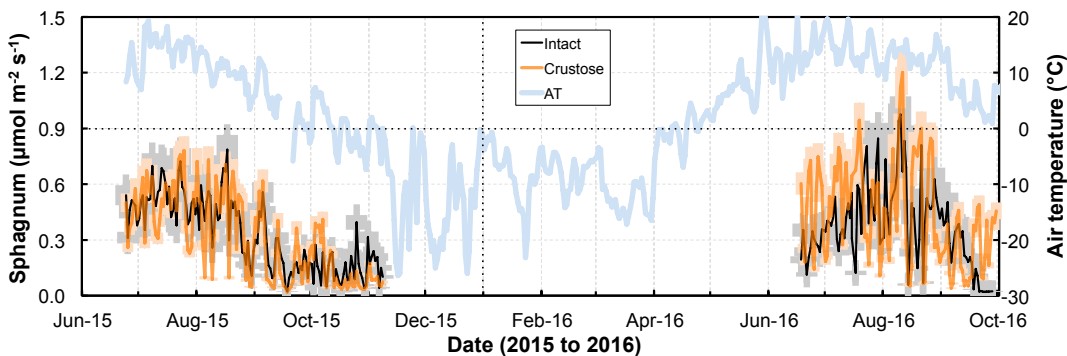

**Figure 3**. Temporal variations in mean daily soil $CO_2$ effluxes with standard deviation (95 %

confidence level) and ambient temperature at intact and crustose sphagnum moss colonies during

the observation periods of 2015 and 2016.





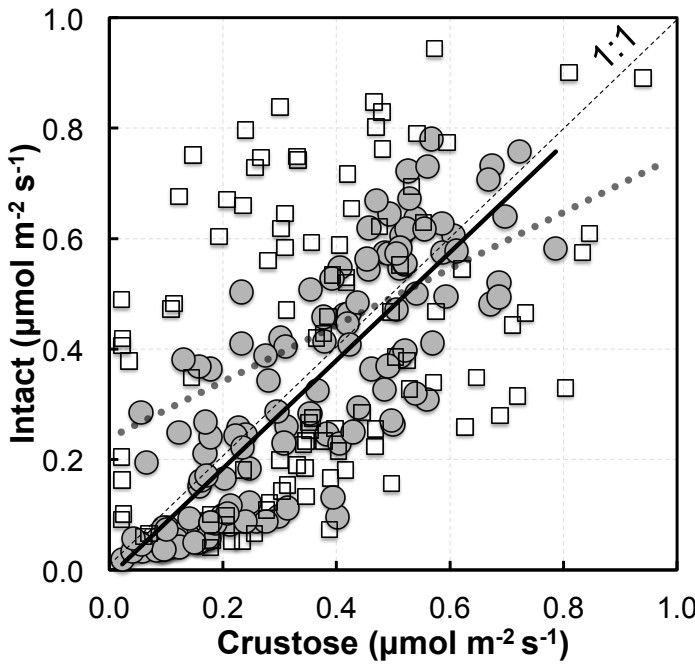

2  **Figure 4**. Responses from soil $CO_2$ effluxes at intact to crustose sphagnum moss during a) 2015

3  (circles) and b) 2016 (squares). The thin dotted line indicates a 1:1 line. Correlation curves for

4  2015 and 2016 are shown by solid and dotted lines, respectively.





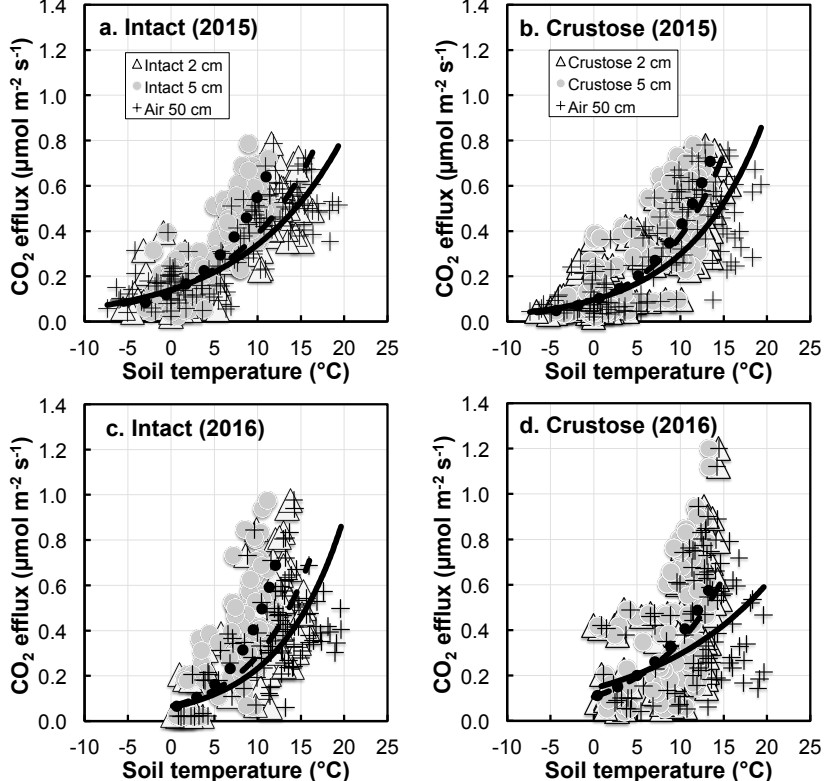

**Figure 5**. Responses from mean daily soil $CO_2$ effluxes to air temperature (pluses), soil temperature at 2 cm (triangles), and 5 cm (grey circles) below the surface at a) intact and b) crustose for 2015, and c) intact and d) crustose for 2016. Correlation curves for air temperature and soil temperature at 2- and 5-cm depths are shown by solid, dashed, and dotted lines, respectively.





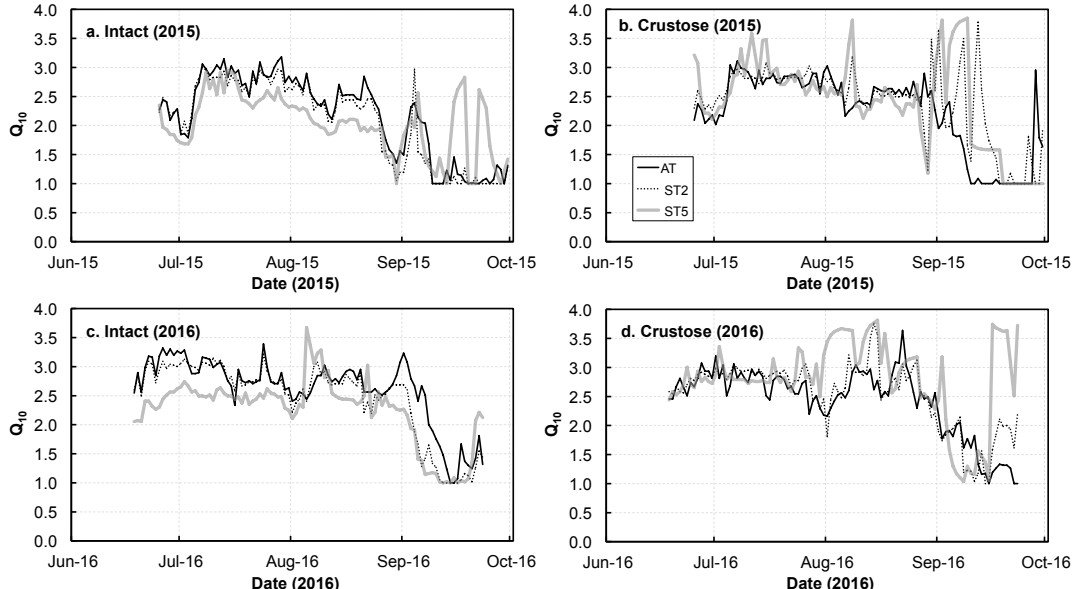

**Figure 6**. Temporal variations in $Q_{10}$ values using equation (6) for air temperature (solid line),
soil temperature at 2 cm (dotted), and 5 cm (grey) below the surface at a) intact and b) crustose
for 2015, and c) intact and d) crustose for 2016. $Q_{10}$ values observed at crustose sphagnum moss
for September of 2015 and 2016 show much wider fluctuation than at intact sphagnum moss.





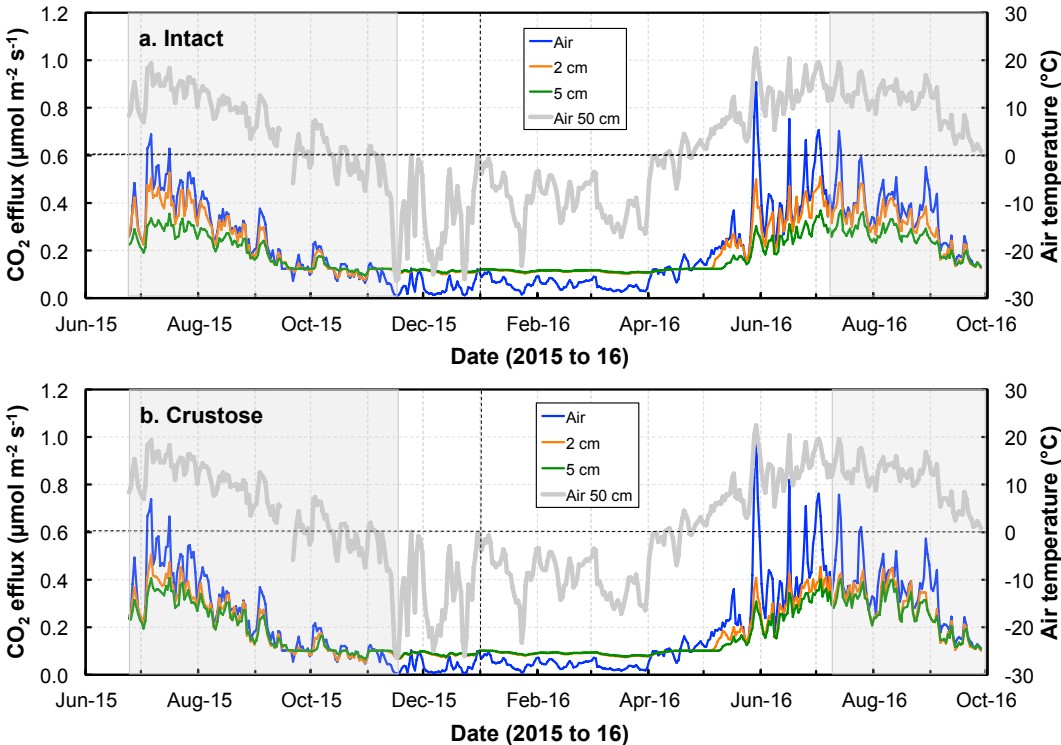

2  **Figure 7**. Temporal variations in soil $CO_2$ efflux simulated by equation (3) and air temperature at

3  a) intact and b) crustose sphagnum moss from June 25, 2015 to September 30, 2016. Shaded

4  columns represent the observation periods of 2015 and 2016.





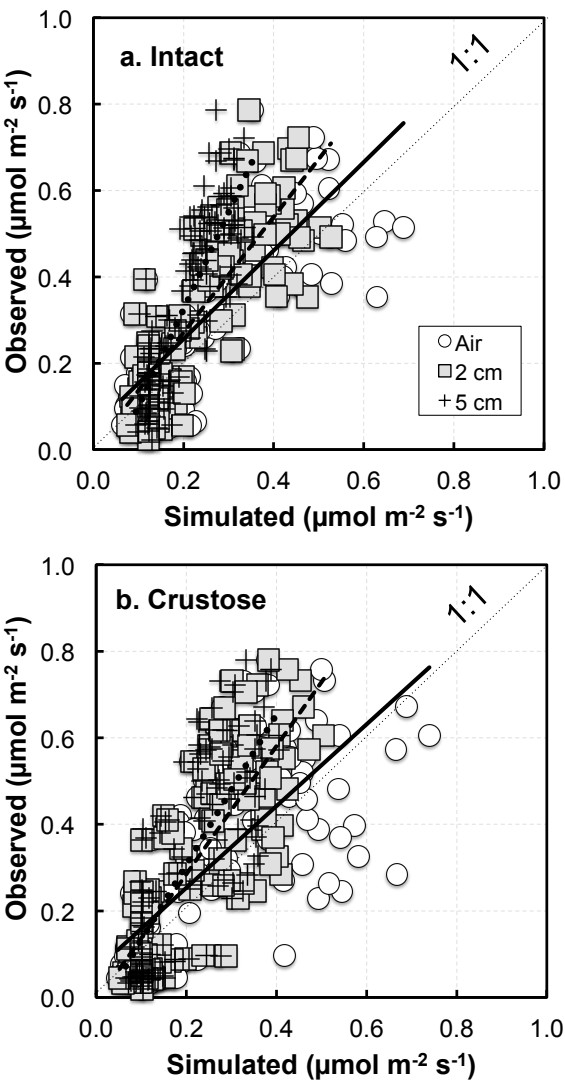

**Figure 8**. Relationship between mean daily observed soil $CO_2$ effluxes ($OSR_{intact}$ and $OSR_{crustose}$)

and simulated soil $CO_2$ efflux (SSR) based on air temperature (AT), soil temperature at 2- (ST2)

and 5-cm (ST5) depths at 1) intact and 2) crustose sphagnum moss. Thin dotted lines indicate a

1:1 line. In Figure a), $OSR_{intact} = 1.02\ SSR_{AT} + 0.05$ ($R^2 = 0.64$) (solid line),

$OSR_{intact} = 1.34\ SSR_{ST2} + 0.01$ ($R^2 = 0.70$) (dashed line), and $OSR_{intact} = 2.24\ SSR_{ST5} - 0.12$

($R^2 = 0.72$) (dotted line), and in Figure b), $OSR_{crustose} = 0.95\ SSR_{AT} + 0.06$ ($R^2 = 0.48$) (solid line),





1    $OSR_{crustose} = 1.48\ SSR_{ST2} + 0.01$   $(R^2 = 0.63)$  (dashed line), and  $OSR_{crustose} = 1.74\ SSR_{ST5} - 0.04$

2    $(R^2 = 0.60)$ (dotted line), respectively.