# Peer review of "Effect of crustose lichen (*Ochrolecia frigida*) on soil CO2"

_Biogeosciences, 2019_

## Referee Comment (RC1) · Anonymous Referee #1 · 8 May 2019

General comments:

This manuscript investigates soil CO2 fluxes in a sphagnum moss community under healthy vs infested by crustose lichen. The authors used novel instrumentation called forced diffusion chamber that allows them to collect high frequency measurement of CO2 fluxes at a microsite during the growing season. From the two growing season observations, the authors show that soil CO2 fluxes in the two microsites are different in a particularly warmer and drier conditions. The authors conclude from these results that higher soil temperature and lower moisture in crustose lichen patches are attributed to enhanced soil CO2 emission. The dataset presented in the manuscript is

quite novel, where the observations focus on the microsite scale measurements of soil $CO_2$ fluxes in healthy sphagnum community and sphagnum community infested with lichen.

Unfortunately, the writing is rather poorly executed, making the manuscript a mere presentation of the measurements. I have several major concerns throughout the manuscript.

First, the way the manuscript is currently written, authors do not provide much insight towards answering 'why' they observed what they observed. Much of the manuscript focuses on methodology of how they came up with modelling yearlong soil $CO_2$ fluxes, which to me could have been a part of supplementary information. The Introduction section goes over a bit far fetched into the biological effects of crustose lichen, but fails to make the link between how lichen infestation affects microclimate or microsite environmental changes to eventually affect soil $CO_2$ fluxes. To me, a novel dataset cannot automatically be granted a publication unless it is written well with a scientific focus. After reading the whole manuscript, I was left with the question 'why is this interesting and important?'. The main conclusion of this study is that the sphagnum and lichen communities showed different soil $CO_2$ fluxes in one of the growing seasons observed and temperature and soil moisture were important parameters in predicting it. However, it is already a widely accepted knowledge that soil respiration largely depends on temperature and moisture. So the question here should be 'what did the lichen infestation do in those microsites to alter temperature and moisture to affect soil $CO_2$ fluxes?'. But the authors fail to provide that link in this manuscript. It is unclear to me whether the reason soil $CO_2$ fluxes in sphagnum vs lichen communities are different is due to sphagnum community affecting environmental conditions or vise versa. What could help the authors to make the manuscript more interesting is to try to focus on hypothesis testing based on the data they have. Perhaps the authors can focus more on answering the question 'why' throughout the manuscript.

Second, the other major concern I have about the methods is the attempt the authors

make to compute running Q10s using the two depths of soil temperature and air temperature. The authors go on in depth showing the fit of Q10 and use this in modelling yearlong soil CO2 fluxes. I do not understand why the authors did this exercise at length. Conventionally, temperature sensitivity of soil respiration, Q10, is computed using soil temperature and when computing Q10 the data are pooled to achieve the best fit of Q10. The authors model yearlong soil CO2 fluxes using this Q10 fit in three different model fits they compute for the two different years' of observations, but I also do not understand why the authors did this exercise when they actually have high frequency measurement of soil CO2 fluxes. What is the purpose of modelling soil CO2 fluxes that show three different sensitivity in temperature when they already have observational data? The modelling should only be used as part of gap filling in this case. The authors should provide better justification of this method.

Third, the authors need to be more careful about the use of language (apart from the use of English as a language) in the manuscript such that the language they use is consistent throughout the manuscript. For instance, one of the most important terms they use in this manuscript is 'sphagnum moss communities', however, several different terms are used throughout (e.g. sphagnum moss regime, crustose lichen patches, sphagnum moss colony, intact sphagnum, sphagnum habitat, and etc.). I suggest consistent use of 'sphagnum dominated' vs 'lichen infested (dominated)' community throughout the manuscript. This is just one example and the authors need more careful usage of terminology throughout. The word 'infected' is used throughout this manuscript to describe lichen dominant sphagnum patches. 'Infected' to describe an invasion of microorganisms and thus the authors should use 'lichen infested or lichen affected' throughout the manuscript. The authors acknowledge that the manuscript has gone through a language check by a native speaker of English, however, I still see language issues throughout the manuscript. I suggest the authors to have the final version edited by a native speaker of English more thoroughly.

Specific comments:

Introduction

- The second paragraph is a very important component of the introduction, where it introduces the logical flow of this study. However, it focuses rather too much on the form and biology of sphagnum and lichen rather than the environmental effects of these two. As the paragraph is unfocused, it makes the logical flow unnatural and weak. The second to last sentence of this paragraph even goes into saying that moss could wither and die, losing its preservation of permafrost. This is a bit of an overstatement making the logical flow weak for this study. Please consider revising the paragraph.

Methods

- There needs a section for data analysis. Please specify what tools are used for data analysis and modelling.

2.1 Sampling Descriptions and Methods

- P6L22: The authors state that the air temperature is measured at 2 m height. This also comes up in P9L12. Then what is Air50 in Table 2 and Figure 7? Please specify this in methods.

2.2 Forced Diffusion (FD) CO2 Efflux Chamber

- Please specify what soil CO2 efflux includes in this study. If surface vegetation (sphagnum/lichen) have been removed, please clarify how lichen infestation may affect soil CO2 efflux.

- Figure1 and associated text (P7L20-26). This is a technical part that does not add much to the science of this manuscript. I suggest moving this part to supplementary information.

2.3 Simulated Soil CO 2 Efflux

- It would be helpful for the readers to understand why the authors compute temperature sensitivity in this study and why this is important. Some background information

and justification of methods used here would be necessary.

3.1 Temporal Variations in Environmental Parameters

- This section can be more focused around how environmental conditions are different under the two different communities investigated and explain why that should be. At this stage, it is rather too lengthy and unfocused. As a result, it is very difficult to grasp what the main findings are.

- P10L7-10 is better suited in the next sub-section.

- P10L22-25: I have a hard time understanding this sentence. It should be revised and perhaps adding a reference would be helpful.

- P11L5-7: This contradicts to the earlier statement 'Peaks in soil moisture during the soil thawing of early May were found at 2- and 5-cm depths in 2015 and 2016 (Figure 2), suggesting the response from soil moisture at 2- and 5-cm depths for intact sphagnum is much more sensitive to soil thawing than at crustose regime.'. Please clarify.

- P11L11-14: Why is it that in 2016 soil moisture was higher in crustose at 2cm depth? The soil temperature and moisture dynamics in relation to lichen dominance should be explained a bit better.

3.2 Seasonal Variations in Soil CO2 Emissions

- P11L23-28: This part largely overlaps with methods and should be moved to methods section.

- P12L11-14: Usually when moisture increases, the rate of organic matter decomposition also increases. Why is it the other way around in this case?

-P12L24: I'm not sure if this is a good comparison as Svalbard soil is very low in soil C compared to AK.

3.3 Sensitivity of Soil CO2 Emissions to Temperature and Soil Moisture

- P13L16: The authors discuss seasonal dependence of soil $CO_2$ efflux here. I do not understand this explanation. The only two environmental variable measured in this study are temperature (at various depths) and soil moisture. So what is the seasonality that regulates soil $CO_2$ efflux in this case? Usually seasonal dependence of ecosystem C exchange is due to physiological changes of vegetation through the season or temperature dependence of respiration with season. In this study, photosynthesis does not come into play and the authors tease out temperature sensitivity in this section, but then what is the seasonal dependence are they referring to? Please clarify.

- P14L7-9: This is also a key explanation the authors keep referring to. I am curious why this is. It would be important to link theories with observations in this case. Otherwise, one of the most important support would largely remain as part of speculation.

- P16 last paragraph: The authors are discussing the usefulness of using FD chambers in this paragraph, but I think it is a bit too far fetched from the main point of the study. Please consider making the final part of the discussion rather focused on the main point of the study.

Technical corrections:

- P3L3: Either 'in time and space' or 'on temporal or spatial scales'

- P3L5-8: This only applies to high latitude ecosystems. Please specify.

- P4L16-19: Please revise this sentence.

- P5L21-23: This sentence already appears in the Introduction section. Please remove.

- P5L24: ecosystem 'dominated by'

- P6L6: these should be mean annual temperature and mean annual precipitation

- P6L11: Is 'average ambient temperature' air temperature? Please clarify.

- P7L17: Please specify that this is due to loss of power.

- P9L7: equation (6) should be (5) instead. Throughout the manuscript, equation (6) is referred to. This needs to be revised.

- P9L18: Soil temperature is 'higher'. This should be consistent throughout.

- P11L1: a sharp jump 'in'.

- P11L16: 'These changes' should be 'The changes'.

- P13L17: Please clarify whether this is combined effects or not.

- P13L21: temperature is 'the' most significant

- P13L28: either 'in' or 'during' August

- P14L12-13: Q10 values at . . . Delete this sentence.

- P16L10-12: This sentence is very difficult to understand and grammatically incorrect. Please revise.

- P16: delete '-measurement' from 'soil CO2 efflux-measurement'.

- P16L21: Please clarify why sunny sky matters. This is winter measurements we're referring to.

- Table1: I do not think this table is very useful. It largely overlaps with the information shown in Figure 2&3. Please consider making it a supplementary information.

- Figure2: Delete the small a) inside the figure. The lines(solid/dotted) of Crustose T and M are easy to identify as they are different colours, but Intact T and M are very difficult to distinguish. Please consider using a different colour for one of them. To me, soil temperature and moisture can both be effect variables and response variables at the same time in this study. This means that soil T and M are the two variables that affects CO2 efflux, but at the same time, they can be affected by the presence of lichen. Therefore, this figure should include air temperature and rainfall data and the description of results should focus around how soil T and M change under variation of

air T and rainfall and how intact and crustose moss affect soil T and M during these events (rainfall) and why that is.

- Figure3: The left y axis should be 'CO2 efflux'. What are the different colours for SD (pink and light purple)? Can the colours (or line form) for two different communities consistent throughout the manuscript? - Figure4: Make a legend indicating dotted and solid lines.

---

## Referee Comment (RC2) · Anonymous Referee #2 · 31 May 2019

The paper by Kim et al. presents an interesting premise: that crustose lichen may affect the CO$_2$ flux of *Sphagnum* moss, and that this infection may also affect the stability of the permafrost beneath. They argue this by presenting data from two flux chambers in a patch of *Sphagnum* moss in western Alaska. This kind of research is valuable, since not much is known on how the spread of lichens influences CO$_2$ fluxes, but unfortunately this study falls short on too many fronts to make meaningful conclusions about this phenomenon. I don't think that the presented data convincingly show that there is a strong effect of crustose lichen on the CO$_2$ flux from these ecosystems.

To begin with the study setup: it's commonly known that CO$_2$ fluxes vary strongly spatially, and it's therefore advisable to use multiple measurements within each vegetation type to reliably determine whether two vegetation types exhibit different CO$_2$ fluxes. The same goes for soil moisture and soil temperature. This experiment, however, uses only one chamber in healthy *Sphagnum* and one in crustose-infected *Sphagnum*. The results then show minute differences between the two, which the authors extrapolate to say something about CO$_2$ fluxes between infected and non-infected *Sphagnum* in general. But without knowing what the variation within each group is, we don't know whether the differences between infected and non-infected *Sphagnum* are meaningful.

The authors claim that the two are different, based on a one-way ANOVA, but this statistical method is not suitable for this study. A one-way ANOVA is used to show whether two groups are taken from the same population by studying the variance between and among groups. In this study, we don't have two groups. Just two time series of repeated measurements. In this situation, a one-way ANOVA is not applicable since the repeated measures are not independent.

With just two measurement locations, it's not possible to show that the two populations from which these measurements were taken (intact and infected *Sphagnum*) exhibit statistically different fluxes since we don't know the variation within each group. In any case, the differences are very small. Visually, it appears that the only period where there are clear differences is for two weeks in June 2016 but the overlap between the two is huge for the rest of the time, which shows that more samples from each group would be required to argue that a difference exists.

Furthermore, the authors claim that their study shows that the spread of crustose lichen would lead to the rapid degradation of permafrost, but not a single measurement of active layer depth is shown in this study. Actually, at a depth of 2 cm, temperatures are lower under the crustose lichen. This is unsurprising, since the photo of the field plots shows that these lichens are completely white, and therefore have a high albedo. This means that a lot of sunlight is reflected, which would actually cool the surface and prevent permafrost degradation. This important property of these lichens is not

mentioned in the paper, and the conclusion that these would lead to rapid permafrost degradation is unsupported.

The presentation of the paper, unfortunately, is also lacking. The writing is often confusing (despite a language check) and many statements are not well-supported by either the data or a citation to another study. The authors try to solve some of the problems caused by the limited data by applying a model, but this shows a poor performance and is subsequently extended to the full winter, a time period on which it was not tested. It's better to focus on the measurements themselves instead of using an imperfect model to draw conclusions.

Overall, I think it's a pity the authors did not do a better job because the data itself is truly interesting. But too many questions remain. For example: the flux measurements are only soil respiration, not net ecosystem exchange. Perhaps the growth of crustose lichen compensates for the loss of carbon from the infected *Sphagnum*? Unfortunately, due to the flaws in the study setup and analysis we are not closer to understanding whether crustose lichens do actually affect the $CO_2$ flux of *Sphagnum* mosses.

More specific comments:

Page 4, line 21-23: this statement is essential to the premise of this paper, but it's not supported by a citation.

Page 4, line 27-30: this is a very basic statement but for some reason the authors need to cite 13 studies including 6 by the main author himself! One citation would suffice.

Page 6, lines 2-3: this paper does not specify which species of *Sphagnum* the measurements are done on. Judging from the photo, I assume it's *Sphagnum fuscum*?

Page 6, line 22: has this sensor been calibrated for moss? It's normally only calibrated for mineral soil (which *Sphagnum* certainly isn't).

Page 8, line 3: this makes no sense. You estimated $CO_2$ flux sensitivity from the exponential relationship between air and soil temperature?

Page 9, line 3: air temperature at 0.5 m, but you measure at 2 m!

Section 3: Why are the results and discussion combined? These should be separated in two sections.

Page 10, line 24: what do you mean with 'forfeiture' in this context?

Page 10, line 26: indicate where this is shown in the figure.

Page 11, line 2: how were these thawing rates calculated?

Page 11, line 5: a thawing rate of 0 cm/day?

Page 11, line 9-10: the sudden drop in soil moisture (and the sudden rise in spring) are probably due to the fact that your moisture sensor doesn't work below 0° C. This is clear from your temperature sensor. Moisture data from days with temperatures below freezing should not be used.

Page 11, line 16-21: these snow depth measurements appear to be from a different location, judging from the vegetation. Why not point a timelapse camera at your plots so you know when the snow melted there, rather than at another place which may not be representative of your measurement location?

Page 11, line 11-14: it is pure speculation to say that this is due to a hotter and drier environment. Again, soil temperatures at 2 cm are lower in the crustose lichen location. Soil moisture is also regularly higher under the crustose lichen. Besides, there is no large difference in 2015 despite similar differences in moisture.

Page 14, line 8-9: this relation with soil moisture is not shown in this study.

Page 15, line 8: it's commonly known that air temperature governs soil temperature. There's no need to cite yourself twice to support that statement.

Page 17, line 17: the data presented in this paper do not show a loss of ecological and thermal functions.

Page 17, line 27-28: by only measuring soil respiration, rather than net ecosystem exchange, it's impossible to say whether shriveled *Sphagnum* moss is a source of $CO_2$ to the atmosphere.

---

## Author Comment (AC1) · 22 Nov 2019

Point-by-point response to Referee's comments

I appreciate the invaluable comments from the Biogeosciences Editorial Office regarding the improvement of this manuscript through careful revision.

\*\*\*

BG-2019-121 (RED color)

"Effect of crustose lichen (Ochrolecia frigida) on soil CO2 efflux in a sphagnum moss community over western Alaska tundra" by Kim, Park and Lee

[Figure]

For clarity, see the Referee #1 in the corrected word file (BG-2019-121-RC1.docx).

Also, I have corrected the manuscript according to the professional English-language editor (Mr. Nate Bauer) of the University of Alaska Fairbanks.   Referee #1

This manuscript investigates soil $CO_2$ fluxes in a sphagnum moss community under healthy vs infested by crustose lichen. The authors used novel instrumentation called forced diffusion chamber that allows them to collect high frequency measurement of $CO_2$ fluxes at a microsite during the growing season. From the two growing season observations, the authors show that soil $CO_2$ fluxes in the two microsites are different in a particularly warmer and drier conditions. The authors conclude from these results that higher soil temperature and lower moisture in crustose lichen patches are attributed to enhanced soil $CO_2$ emission. The dataset presented in the manuscript is quite novel, where the observations focus on the microsite scale measurements of soil $CO_2$ fluxes in healthy sphagnum community and sphagnum community infested with lichen. Unfortunately, the writing is rather poorly executed, making the manuscript a mere presentation of the measurements. I have several major concerns throughout the manuscript. »> I appreciate your invaluable advices and comments on this manuscript, and will correct/rewrite/add text for reader comprehension about the scientific importance.

First, the way the manuscript is currently written, authors do not provide much insight towards answering 'why' they observed what they observed. Much of the manuscript focuses on methodology of how they came up with modelling yearlong soil $CO_2$ fluxes, which to me could have been a part of supplementary information. The Introduction section goes over a bit far fetched into the biological effects of crustose lichen, but fails to make the link between how lichen infestation affects microclimate or microsite environmental changes to eventually affect soil $CO_2$ fluxes. To me, a novel dataset cannot automatically be granted a publication unless it is written well with a scientific focus. After reading the whole manuscript, I was left with the question 'why is this interesting and important?'. The main conclusion of this study is that the sphagnum

and lichen communities showed different soil $CO_2$ fluxes in one of the growing seasons observed and temperature and soil moisture were important parameters in predicting it. However, it is already a widely accepted knowledge that soil respiration largely depends on temperature and moisture. So the question here should be 'what did the lichen infestation do in those microsites to alter temperature and moisture to affect soil $CO_2$ fluxes?'. But the authors fail to provide that link in this manuscript. It is unclear to me whether the reason soil $CO_2$ fluxes in sphagnum vs lichen communities are different is due to sphagnum community affecting environmental conditions or vise versa. What could help the authors to make the manuscript more interesting is to try to focus on hypothesis testing based on the data they have. Perhaps the authors can focus more on answering the question 'why' throughout the manuscript »> I may have been unclear about the hypotheses/questions about the difference between $CO_2$ emissions from intact and lichen infested sphagnum moss communities. As you suggest, the text may fail to make the link between how lichen infection affects microclimate or microsite environmental changes to eventually affect soil $CO_2$ emission.

»> I added the paragraphs in L23 of P4 and rewrote L23-25 of P4, as suggested.

»> These are the direct biological effect of lichen infestation on intact sphagnum moss. The indirect thermal and biogeochemical effects of lichen-infested sphagnum moss occur under the microclimate conditions of in soil temperature and soil moisture, decomposition of soil organic matter, and subsequently soil $CO_2$ emission at the microsite, compared to the intact sphagnum moss. As a result, uncertainty still remains about the ecological and biogeochemical roles of crustose lichen-infested sphagnum moss, despite of gradual widespread occurrence of lichen infestation in the recently warming Subarctic and Arctic.

»> Here we investigated the difference in soil carbon emission from intact and crustose lichen-infested sphagnum moss in a microsite tundra ecosystem during the growing season

»> I also added sentences before L9 of P5 and corrected L9-14 of P5, as suggested.

»> We propose the important question of how crustose lichen infestation affects intact sphagnum functions, alters soil temperature and moisture and governs soil $CO_2$ emission. This study focuses on the biogeochemical effects of crutose lichen-infested sphagnum moss at the microsite, in response to microclimate changes in soil temperature and moisture, as well as in soil $CO_2$ emission. In response, the purposes of this study are to 1) determine the environmental drivers (e.g., soil temperature and moisture) resolving soil $CO_2$ emissions in intact and crustose lichen infested sphagnum moss of the tundra ecosystem in western Alaska; 2) estimate soil $CO_2$ emission in these microsites by continuous forced diffusion (FD) chamber system during the growing seasons of 2015 and 2016; and 3) assess the contributions from seasonally snow-covered- and snow-free-period carbon toward the simulated annual carbon budget, based on in-situ temperature and snow depth.

Second, the other major concern I have about the methods is the attempt the authors make to compute running Q10s using the two depths of soil temperature and air temperature. The authors go on in depth showing the fit of Q10 and use this in modelling yearlong soil $CO_2$ fluxes. I do not understand why the authors did this exercise at length. Conventionally, temperature sensitivity of soil respiration, Q10, is computed using soil temperature and when computing Q10 the data are pooled to achieve the best fit of Q10. The authors model yearlong soil $CO_2$ fluxes using this Q10 fit in three different model fits they compute for the two different years' of observations, but I also do not understand why the authors did this exercise when they actually have high frequency measurement of soil $CO_2$ fluxes. What is the purpose of modelling soil $CO_2$ fluxes that show three different sensitivity in temperature when they already have observational data? The modelling should only be used as part of gap filling in this case. The authors should provide better justification of this method. »> I have described the estimation of Q10 regarding 1) the gap-filling of the early growing season of 2016 and the contribution of soil $CO_2$ emission (equations 2 and 3) as suggested, and 2) the temporal variation of Q10 (e.g., temperature sensitivity of soil CO2 emissions in response to ambient and soil temperatures at intact and crustose lichen-infested sphagnum moss during the growing season (equations 4 and 5).

»> I agree with the comments about the analysis of high-resolution CO2 flux-measurement with FD chamber. However, because soil CO2 emissions in intact and crustose lichen infested sphagnum moss represent only a subtle difference, despite high-frequency measurements of soil CO2 fluxes, I have tried to assess the biogeo-chemical effect of crustose lichen-infested sphagnum moss. Furthermore, although the temperature is a more significant driver in regulating soil CO2 emissions in intact sphagnum than in crustose lichen-infested sphagnum moss, soil CO2 emission in crustose lichen infested sphagnum is only a little higher than in intact sphagnum moss. And so I calculated annually simulated soil CO2 emission based on the equations 2 and 3.

»> Because I measured high-frequency measurement of soil CO2 emissions, I then computed the temporal variation in Q10 in response to atmospheric and soil tempera-tures in intact and crustose lichen infested sphagnum moss microsites during the two growing seasons of 2015 and 2016. I added the sentence to the end of L24 of P8, as suggested.

»> using the following two models for the temporal variation of Q10 on the atmospheric and soil temperatures (Ueyama et al., 2014):

»> I have plotted the relationship between soil CO2 efflux and soil moisture in intact and crustose lichen infested sphagnum moss. However, because the relationships were weaker, I did not report the Figure in the text.

Third, the authors need to be more careful about the use of language (apart from the use of English as a language) in the manuscript such that the language they use is consistent throughout the manuscript. For instance, one of the most important terms they use in this manuscript is 'sphagnum moss communities', however, several differ-ent terms are used throughout (e.g. sphagnum moss regime, crustose lichen patches,

sphagnum moss colony, intact sphagnum, sphagnum habitat, and etc.). I suggest consistent use of 'sphagnum dominated' vs 'lichen infested (dominated)' community throughout the manuscript. This is just one example and the authors need more careful usage of terminology throughout. The word 'infected' is used throughout this manuscript to describe lichen dominant sphagnum patches. 'Infected' to describe an invasion of microorganisms and thus the authors should use 'lichen infested or lichen affected' throughout the manuscript. The authors acknowledge that the manuscript has gone through a language check by a native speaker of English, however, I still see language issues throughout the manuscript. I suggest the authors to have the final version edited by a native speaker of English more thoroughly. »> I corrected 'intact sphagnum moss' and 'crustose lichen-infested sphagnum moss' throughout the text for consistent terminology, as suggested. I deleted words such as 'regime', 'patches', 'colony' and 'habitat' except for 'community' in the text, as suggested.

»> Also, A native speaker of English has thoroughly checked the final version, as suggested.

Introduction - The second paragraph is a very important component of the introduction, where it introduces the logical flow of this study. However, it focuses rather too much on the form and biology of sphagnum and lichen rather than the environmental effects of these two. As the paragraph is unfocused, it makes the logical flow unnatural and weak. The second to last sentence of this paragraph even goes into saying that moss could wither and die, losing its preservation of permafrost. This is a bit of an overstatement making the logical flow weak for this study. Please consider revising the paragraph. »> I deleted L21 to 23 of P4, and added text for the biological and biogeochemical effects of crustose lichen-infested sphagnum moss in L21 of P4, as suggested.

Methods - There needs a section for data analysis. Please specify what tools are used for data analysis and modelling. »> Section 2.3 is for data analysis and modeling. I have added the title of section 2.3 and two phases of L20 and L24 of P8, as follows.

»> 2.3 Simulated Soil CO2 Efflux and Temporal Variation of Q10

»> L20 of P8: for the seasonal variation of soil CO2 efflux in response to temperature, were calculated as:

»> L24 of P8: using the following two models for the temporal variation of Q10 on the atmospheric and soil temperatures (Ueyama et al., 2014):

»> Could you please let me know your specific comments about tools for data analysis and modeling?

2.1 Sampling Descriptions and Methods - P6L22: The authors state that the air temperature is measured at 2 m height. This also comes up in P9L12. Then what is Air50 in Table 2 and Figure 7? Please specify this in methods. »> It is 2 m in height. I corrected it in Figure 7 and Table 2 in the text, as suggested.

2.2 Forced Diffusion (FD) CO2 Efflux Chamber - Please specify what soil CO2 efflux includes in this study. If surface vegetation (sphagnum/lichen) have been removed, please clarify how lichen infestation may affect soil CO2 efflux. »> Because the crustose lichen was infested on the surface sphagnum moss, I could not remove the surface plants, with Risk et al. (2011) for reference.

- Figure1 and associated text (P7L20-26). This is a technical part that does not add much to the science of this manuscript. I suggest moving this part to supplementary information. »> I wholly understand the issue as suggested. However, this is very important information, because measuring time frequency could provide comparisons between automatic open-top chamber (3.75-m resolution) and eddy covariance tower methods (30-m frequency) at the site. My colleagues have used automatic chamber and eddy covariance tower systems. However, they did not obtain satisfactory data due to the frequent malfunction of power supply systems. Unfortunately, I could not compare my data with theirs. Therefore, I wish to introduce Figure 1 and the associated paragraph in the text for additional study in the future.

2.3 Simulated Soil CO2 Efflux - It would be helpful for the readers to understand why the authors compute temperature sensitivity in this study and why this is important. Some background information and justification of methods used here would be necessary. »> I added text in L2 of P8 as follows, as suggested.

»> Based on the recently increasing temperature of Alaska (Bieniek et al., 2014), crustose lichen may have recently infested over intact sphagnum moss on a microsite scale (see Figure S1). Furthermore, the increase in temperature may have triggered new soil CO2 emissions from 2008 to 2015 in the tundra ecosystem of Alaska, which would represent a significant feedback toward further warming (Euskirchen et al., 2017). Here we investigated how soil CO2 efflux in crustose sphagnum moss responds to microclimate changes in temperature within a microsite.

3.1 Temporal Variations in Environmental Parameters - This section can be more focused around how environmental conditions are different under the two different communities investigated and explain why that should be. At this stage, it is rather too lengthy and unfocused. As a result, it is very difficult to grasp what the main findings are. »> I added two paragraphs in L7 of P10 as follows, as suggested.

»> Further, the mean growing season differences in soil temperature between the two growing seasons of 2015 and 2016 were 1.51 °C at 2 cm depth and 1.40 °C at 5-cm depth for intact sphagnum moss, respectively. At crustose lichen-infested sphagnum moss, the differences were 1.51 °C at 2-cm depth, and 1.46 °C at 5-cm depth, respectively. This results from the higher atmospheric temperature in 2016 relative to 2015.

»> The mean growing season differences in soil temperature between intact and crustose lichen-infested sphagnum moss were -0.47 and 0.98 °C at 2-cm depth in 2015 and -0.47 and 1.04 °C at 5-cm depth in 2016, respectively. Crustose lichen-infested sphagnum moss consists of dried dead sphagnum, and loses water-holding capacity. Because the heat capacity at surface intact sphagnum moss is higher than for crustose

lichen-infested sphagnum, soil temperature at 2-cm depth living sphagnum is relatively higher than for crustose lichen-infested sphagnum. Soil temperature at 5-cm depth for intact sphagnum is much lower than for the crustose sphagnum moss community.

- P10L7-10 is better suited in the next sub-section. »> I moved L7-10 of P10 to L23 of P11 to after the correction as follows, as suggested.

»> The increase in soil $CO_2$ emission during the growing season of 2016 relative to 2015 is thought to be causally connected to higher temperatures in air and soil, as well as lower soil water content in 2016 than 2015—key drivers in regulating soil $CO_2$ production and emission to the atmosphere (Xu and Qi, 2001; Davidson and Janssens, 2006; Kim et al., 2014b; 2016).

- P10L22-25: I have a hard time understanding this sentence. It should be revised and perhaps adding a reference would be helpful. »> I revised L22-25 of P10 as follows, as suggested.

»> Differences in soil moistures between 2 and 5 cm depths for crustose infested sphagnum moss relative to intact sphagnum moss did not appear remarkable, reflecting that the infestation of crustose lichen (O. frigida) on intact sphagnum moss may induce the loss of physiological and ecological functions in sphagnum moss (Hahn et al., 1993; Lange et al., 1996). Therefore, crustose lichen-infested sphagnum moss loses its unique water holding capacity.

- P11L5-7: This contradicts to the earlier statement 'Peaks in soil moisture during the soil thawing of early May were found at 2- and 5-cm depths in 2015 and 2016 (Figure 2), suggesting the response from soil moisture at 2- and 5-cm depths for intact sphagnum is much more sensitive to soil thawing than at crustose regime'. Please clarify. »> I corrected L5-7 of P11 as follows, as suggested.

»> There was no difference in thawing rate between 2- and 5-cm depths for the crustose lichen-infested sphagnum community in early spring, reflecting that the inherent

higher water-holding capacity for intact sphagnum was completely lost, as the former sphagnum moss was dead.

- P11L11-14: Why is it that in 2016 soil moisture was higher in crustose at 2cm depth? The soil temperature and moisture dynamics in relation to lichen dominance should be explained a bit better. »> I have completely checked data for July-August hourly variations in soil temperature (above) and moisture (below) at intact and crustose lichen-infested sphagnum moss, as follows.

»> However, I could not find any clues about why soil moisture at 2-cm depth of crustose lichen-infested sphagnum moss relative to 5-cm depth. Hence, we must have additional observations, such as soil moisture and temperature at several centimeter depths, for the ascertainment of soil temperature and moisture dynamics for the unusual results. I appreciate your invaluable comments and suggestions for future work.

3.2 Seasonal Variations in Soil CO2 Emissions - P11L23-28: This part largely overlaps with methods and should be moved to methods section. »> I deleted L24-25 of P11 and moved L26-28 of P11 to L26 of P7 in the end of session 2.2, as suggested.

- P12L11-14: Usually when moisture increases, the rate of organic matter decomposition also increases. Why is it the other way around in this case? »> According to Kim et al. (2014b), $CO_2$ efflux tended to increase with an increase in soil moisture when soil moisture value was at the optimum 0.228 m3 m-3 in the tundra ecosystem. $CO_2$ efflux decreases over the optimum soil moisture (Kim et al., 2014b). Davidson et al. (1998) also reported a correlation between soil water content and $CO_2$ efflux in different drainage classes. $CO_2$ efflux increased when soil water content was less than 0.2 m3 m-3; on the other hand, higher soil moisture resulted in a decrease in $CO_2$ efflux (see Fig. 7, Davidson et al., 1998).

»> The two papers are listed in the references.

-P12L24: I'm not sure if this is a good comparison as Svalbard soil is very low in soil

C compared to AK. »> I agreed with your comments; however, it is difficult to find references despite the difference of soil organic carbon at the two sites.

3.3 Sensitivity of Soil CO2 Emissions to Temperature and Soil Moisture - P13L16: The authors discuss seasonal dependence of soil CO2 efflux here. I do not understand this explanation. The only two environmental variable measured in this study are temperature (at various depths) and soil moisture. So what is the seasonality that regulates soil CO2 efflux in this case? Usually seasonal dependence of ecosystem C exchange is due to physiological changes of vegetation through the season or temperature dependence of respiration with season. In this study, photosynthesis does not come into play and the authors tease out temperature sensitivity in this section, but then what is the seasonal dependence are they referring to? Please clarify.

- P13L16: The authors discuss seasonal dependence of soil CO2 efflux here. I do not understand this explanation. »> Soil CO2 efflux depends on seasonal changes in ambient and soil temperature, with regard to your comments. If so, I do not doubt it based on other references.

»> I have noted this in L17 of P13, in which ambient and soil temperatures explain 60 % of growing season variability in soil CO2 efflux at intact and crustose lichen-infested sphagnum moss communities.

»> Due to the temperature sensitivity of soil CO2 efflux, I made comparisons between observed and simulated CO2 efflux (Figure 8), computed using equations 2 and 3, based on equation 1.

»> Therefore, soil CO2 efflux is determined by seasonal changes in temperature. Soil moisture does not regulate soil CO2 efflux (not shown), and the relationship between soil CO2 efflux and soil moisture is much weaker (> R2 of 0.15).

- P14L7-9: This is also a key explanation the authors keep referring to. I am curious why this is. It would be important to link theories with observations in this case. Otherwise,

one of the most important support would largely remain as part of speculation. »>
I corrected and added text about the relationship between soil $CO_2$ efflux and soil
moisture to the L8 of P13 after deleting L8-9 of P13, as follows, and did not show them
in the text.

»> Soil moisture acts as a well-known key role in restraining soil $CO_2$ emissions; $CO_2$
flux = 2.61 exp (-8.73×SM2 cm) (R2=0.08) and $CO_2$ flux = 1.68 exp (-6.66×SM5
cm) (R2=0.14) at two depths of intact sphagnum moss, and $CO_2$ flux = 0.81 exp (-
4.01×SM2 cm) (R2=0.03) and $CO_2$ flux = 6.89 exp (-13.1×SM5 cm) (R2=0.12) in
crustose sphagnum moss during two growing seasons of 2015 and 2016, respectively
(not shown).

- P16 last paragraph: The authors are discussing the usefulness of using FD chambers
in this paragraph, but I think it is a bit too far fetched from the main point of the study.
Please consider making the final part of the discussion rather focused on the main
point of the study. »> I agree with your comments, and deleted L18-25 of 16 for the
better comprehension, as suggested.

Technical corrections: - P3L3: Either 'in time and space' or 'on temporal or spatial
scales' »> I corrected to 'in time and space,' as suggested.

- P3L5-8: This only applies to high latitude ecosystems. Please specify. »> Yes, it
applies to high northern latitude ecosystem, as suggested.

- P4L16-19: Please revise this sentence. »> I revised L16-19 of P4 as follows.

»> However, it is not well-known how crustose lichen-infested sphagnum moss affects
intact sphagnum moss, which is commonly distributed over several moss species and
peats in the high Arctic (Gary Laursen; personal communication)

- P5L21-23: This sentence already appears in the Introduction section. Please remove.
»> I deleted L21-23 of P5, as commented.

- P5L24: ecosystem 'dominated by' »> I added to 'dominant by' L21-23 of P5.

[Figure]

- P6L6: these should be mean annual temperature and mean annual precipitation »> I think the mean annual temperature is correct, but the precipitation is total annual amount (ca, 400 mm), as follows.

- P6L11: Is 'average ambient temperature' air temperature? Please clarify. »> I changed 'ambient temperature' to 'air temperature' throughout the text, as suggested.

- P7L17: Please specify that this is due to loss of power. »> I rewrote the following sentence, as suggested.

»> However, we could not determine the winter season CO2 efflux during the observation periods of 2015 and 2016 due to the loss of electric power.

- P9L7: equation (6) should be (5) instead. Throughout the manuscript, equation (6) is referred to. This needs to be revised. »> I changed '(6)' to '(5)' throughout the text, as suggested.

- P9L18: Soil temperature is 'higher'. This should be consistent throughout. »> I changed 'greater' to 'higher' throughout the text, as suggested.

- P11L1: a sharp jump 'in'. »> I changed 'of' to 'in,' as suggested.

- P11L16: 'These changes' should be 'The changes'. »> I changed 'These' to 'The,' as suggested.

- P13L17: Please clarify whether this is combined effects or not. »> I rewrote L17-19 of P13, as follows.

»> Average temperatures in air and soil at 2 and 5 cm depths elucidates 63 % and 45 % of variability in soil CO2 effluxes at intact sphagnum moss during the two growing seasons of 2015 and 2016, respectively. Also, average temperature in air and soil at 2 and 5 cm depths of crutose lichen-infested sphagnum moss explains 50 % and 35 % of variability in soil CO2 effluxes during the two growing seasons. Hence the sensitivity of soil CO2 effluxes to temperature in 2016 was much lower than in 2015.

**BGD**

- P13L21: temperature is 'the' most significant »> I changed 'a' to 'the,' as suggested.

- P13L28: either 'in' or 'during' August »> I changed 'for' to 'in,' as suggested.

- P14L12-13: Q10 values at . . . Delete this sentence. »> I deleted L12-13 of P14, as suggested.

- P16L10-12: This sentence is very difficult to understand and grammatically incorrect. Please revise. »> I revised L10-12 of P16 as follows.

»> The contribution from winter carbon emissions was 20.0, 30.5, and 20.0 % of annual soil $CO_2$ effluxes simulated by temperatures of air and soil 2- and 5-cm depths of intact sphagnum moss, and 16.2, 28.4, and 30.4 % of simulated annual soil $CO_2$ effluxes in crustose lichen-infested sphagnum moss. Although this study did not conduct winter soil $CO_2$ efflux due to loss of power, simulated soil $CO_2$ effluxes during non-growing seasons were within the ranges observed in the Subarctic and Arctic.

- P16L13: delete '-measurement' from 'soil $CO_2$ efflux-measurement'. »> I corrected '-measurement,' as suggested.

- P16L21: Please clarify why sunny sky matters. This is winter measurements we're referring to. »> I deleted L18-25 of P16, as previously described, as suggested.

- Table1: I do not think this table is very useful. It largely overlaps with the information shown in coFigure 2&3. Please consider making it a supplementary information. »> Your comments are appreciated. However, Table 1 also shows important data despite daily temporal variations in environmental and soil $CO_2$ efflux, as shown in Figures 2 and 3. I think Table 1 provides the readers with information about the monthly ratio of crustose-infested lichen to intact sphagnum moss for soil $CO_2$ efflux, soil temperature, and soil moisture.

»> If possible, I do not want to remove Table 1 in the text relative to the supplementary table.

- Figure2: Delete the small a) inside the figure. The lines (solid/dotted) of Crustose T and M are easy to identify as they are different colours, but Intact T and M are very difficult to distinguish. Please consider using a different colour for one of them. To me, soil temperature and moisture can both be effect variables and response variables at the same time in this study. This means that soil T and M are the two variables that affects $CO_2$ efflux, but at the same time, they can be affected by the presence of lichen. Therefore, this figure should include air temperature and rainfall data and the description of results should focus around how soil T and M change under variation of air T and rainfall and how intact and crustose moss affect soil T and M during these events (rainfall) and why that is. »> Figure 2 consists of two panels of a) 2-cm (upper) and b) 5-cm (lower).

»> I changed the different colors of T and M for better comprehension, and added heavy rainfall events with downward arrow marks to Figure 2(a) during two growing seasons, as suggested.

»> I re-plotted Figure 2(a) including air temperature. Air temperature is synchronized with soil temperature in intact and lichen-infested sphagnum moss, which may be hard to demonstrate alongside three temperatures for air and soil in the modified Figure 2(a), as follows.

»> I have also added heavy rainfall events (e.g., downward grey arrows) to Figure 2(a), which affects the response from soil moisture, as commented.

»> Soil moisture in intact sphagnum moss is more sensitive to heavy rainfall events than lichen-infested sphagnum moss, suggesting that dead sphagnum moss infested by crustose lichen lost its water retaining ability, compared to healthy sphagnum moss. Soil temperature is a significant driver in regulating soil $CO_2$ efflux in lichen-infested sphagnum moss relative to soil moisture. On the other hand, soil temperature and moisture are important parameters for modulating soil $CO_2$ efflux, as is well known.

- Figure3: The left y axis should be 'CO2 efflux'. What are the different colours for

SD (pink and light purple)? Can the colours (or line form) for two different communities consistent throughout the manuscript? »> I changed the y-axis to 'CO2 efflux (umol/m2/s), as suggested. Also, mean $\pm$ SD is solid $\pm$ 95% SD, intact sphagnum is a solid line $\pm$ grey, crustose is an orange solid line, and $\pm$ pale orange.

- Figure4: Make a legend indicating dotted and solid lines. »> I added the legend to Figure 4, as suggested.

Please also note the supplement to this comment:
https://www.biogeosciences-discuss.net/bg-2019-121/bg-2019-121-AC1-supplement.pdf

—————————————————————

[Figure]

**Fig. 1.** Temporal variations in temperature and moisture

[Figure]

**Fig. 2.** Temporal variations in soil CO2 emission and air temperature

[Figure]

**Fig. 3.** Responses from soil CO2 effluxes at intact to crustose sphagnum moss

[revised manuscript text omitted]

---

## Author Comment (AC2) · 22 Nov 2019

Point-by-point response to Referee's comments

I appreciate the invaluable comments from the Biogeosciences Editorial Office regarding the improvement of this manuscript through careful revision.
* * *
BG-2019-121

"Effect of crustose lichen (Ochrolecia frigida) on soil CO2 efflux in a sphagnum moss community over western Alaska tundra" by Kim, Park and Lee

[Figure]

For clarity, see the Referee #2 in the corrected word file (BG-2019-121-RC2-Kim.doc).

Also, I have corrected the manuscript according to the professional English-language editor (Mr. Nate Bauer) of the University of Alaska Fairbanks.   Referee #2

The paper by Kim et al. presents an interesting premise: that crustose lichen may affect the $CO_2$ flux of Sphagnum moss, and that this infection may also affect the stability of the permafrost beneath. They argue this by presenting data from two flux chambers in a patch of Sphagnum moss in western Alaska. This kind of research is valuable, since not much is known on how the spread of lichens influences $CO_2$ fluxes, but unfortunately this study falls short on too many fronts to make meaningful conclusions about this phenomenon. I don't think that the presented data convincingly show that there is a strong effect of crustose lichen on the $CO_2$ flux from these ecosystems.

To begin with the study setup: it's commonly known that $CO_2$ fluxes vary strongly spatially, and it's therefore advisable to use multiple measurements within each vegetation type to reliably determine whether two vegetation types exhibit different $CO_2$ fluxes. The same goes for soil moisture and soil temperature. This experiment, however, uses only one chamber in healthy Sphagnum and one in crustose-infected Sphagnum. The results then show minute differences between the two, which the authors extrapolate to say something about $CO_2$ fluxes between infected and non-infected Sphagnum in general. But without knowing what the variation within each group is, we don't know whether the differences between infected and non-infected Sphagnum are meaningful. »> I fully understand your invaluable comments on the spatiotemporal variations of $CO_2$ effluxes in intact and crutose-lichen infected sphagnum.

»> To infer the difference of $CO_2$ effluxes between intact and infected sphagnum, the environmental parameters (e.g., soil temperature and moisture) are significant drivers in directly influencing two different environments. The quantitative elucidation of these data might be helped the reason in difference of $CO_2$ effluxes from both patches. As Alaska is getting warmer, the extent of crutose lichen will widely spread and wither more

sphagnum moss that is directly associated with the extent and depth of permafrost in the tundra ecosystem.

»> Furthermore, before the deployment of FD (forced diffusion) chamber systems, I have observed Re (ecosystem respiration) and NEE (net ecosystem exchange) in eight patches that are covered by intact sphagnum and crustose lichen infected mosses with portable opaque and transparent chambers for the growing season (June to September) of 2013 and 2014 (Figure S1). As the results, ecosystem respirations in intact and crutose-infected moss patches were 0.94±0.75 and 1.36±0.73 $\mu$mol/m2/s for 2013 (n=28) and 1.92±1.67 and 2.45±1.40 $\mu$mol/m2/s for 2014 (n=60), respectively. It represented distinct differences in spatiotemporal variation between both communities. Then, we selected the representative sites for the monitoring of continuous CO2 efflux in two communities with FD chamber systems during the growing seasons of 2015 and 2016.

»> I added the data to the text (end of L28 of page 6) as your comments for the better understanding of spatial variation in CO2 effluxes, as follows.

»> To select representative intact and crustose infected sphagnum sites, ecosystem respiration (Re) and net ecosystem exchange (NEE) were observed in seven communities with manual opaque and transparent chambers during the growing seasons (June to September) of 2013 and 2014. Mean growing season ecosystem respirations in intact and crutose lichen-infested sphagnum mosses were 0.94 ± 0.75 and 1.36 ± 0.73 $\mu$mol m-2 s-1 for 2013 (n = 28) and 1.92 ± 1.67 and 2.45 ± 1.40 $\mu$mol m-2 s-1 for 2014 (n = 60), respectively. Respiration demonstrated distinct differences in spatiotemporal variation across both communities. We also chose representative sites for monitoring of continuous CO2 efflux across two communities using FD (forced diffusion) chamber systems during the growing seasons of 2015 and 2016.

The authors claim that the two are different, based on a one-way ANOVA, but this statistical method is not suitable for this study. A one-way ANOVA is used to show whether

two groups are taken from the same population by studying the variance between and among groups. In this study, we don't have two groups. Just two time series of repeated measurements. In this situation, a one-way ANOVA is not applicable since the repeated measures are not independent. »> I have two groups: one is intact sphagnum moss and the other is crutose lichen infected sphagnum moss with two FD chamber systems at the representative sites. However, I changed a one-way ANOVA to a t-test in the text, Figures, and Tables, as your comment.

With just two measurement locations, it's not possible to show that the two populations from which these measurements were taken (intact and infected Sphagnum) exhibit statistically different fluxes since we don't know the variation within each group. In any case, the differences are very small. Visually, it appears that the only period where there are clear differences is for two weeks in June 2016 but the overlap between the two is huge for the rest of the time, which shows that more samples from each group would be required to argue that a difference exists. »> As previously described, I have observed $CO_2$ effluxes at the representative sites with two FD chamber systems. It may be difficult to distinguish the visible difference in the seasonal variations of mean daily $CO_2$ effluxes between both communities as shown Figure 3 and mean growing seasons of 2015 and 2016 as R2 pointed out. However, I showed the evident differences in Table 1 (mean monthly $CO_2$ efflux) and Figure 4 (mean daily $CO_2$ efflux) as described in the chapter 3.2 of text.

»> In special, the ratio of infected to intact $CO_2$ efflux during the growing seasons of 2015 and 2016 enables the readers to understand the enhancement of $CO_2$ efflux in crutose lichen infected relative to intact sphagnum moss for 2016. Further, if mean daily $CO_2$ effluxes between both communities have a little difference, I would not present Figure 3. And then, I need to additional work such as winter contribution of $CO_2$ effluxes from infected and intact sphagnum after the improvement of power supply system.

Furthermore, the authors claim that their study shows that the spread of crustose lichen

would lead to the rapid degradation of permafrost, but not a single measurement of active layer depth is shown in this study. Actually, at a depth of 2 cm, temperatures are lower under the crustose lichen. This is unsurprising, since the photo of the field plots shows that these lichens are completely white, and therefore have a high albedo. This means that a lot of sunlight is reflected, which would actually cool the surface and prevent permafrost degradation. This important property of these lichens is not mentioned in the paper, and the conclusion that these would lead to rapid permafrost degradation is unsupported. »> I agreed to your comments on a single measurement of active layer depth for rapid degradation of permafrost. Also, I understand the effect of albedo in white-colored crustose infected sphagnum. As I showed in Figure A1 (b to d), new crustose is much white than aged crustose, and two colonies are co-existed nearby. Therefore, it is really difficult to monitor the changes in albedo from new crustose to old with time on a tiny scale and to install the sensors that are required to lots of efforts.

»> I knew that the infected patches were fissured with time as shown in Figure S1. According to the degeneration of crutose lichen, the surface color was getting darker as shown in Figure S1 (b to d). I showed the full size of Figure S1 (d), as follows.

Figure A. The dotted yellow line denotes the boundary of infected and intact sphagnum. Old crustose (dotted green circle) colonies represent much darker relative to new crustose (dotted green oval) and intact brown sphagnum moss communities. It represents the difference in albedo at each colony.

»> It is remarkable that the cortex of the "host lichen" occasionally turns black at the early contact points with O. frigida. Light microscopical investigations and even scanning electron micrographs do not show any apparent anatomical or morphological changes in the cortex (Gr$\beta$mann and Ott, 2000). So, I can infer the change in color of cortex through the growth processes.

»> Although I have measured the thawing depths at 81 points at an interval of 5 m

within 40 m x 40 m plot, I could not do the thawing depth in crutose infected sphagnum colonies. It is due to close two location and disturbance after the measurement of thawing depth with a fiberglass tile probe (1.0 cm diameter, 150 cm long). Also, I have tried to monitor the temperature profile of active layer and permafrost in crutose infected sphagnum patches; however, it is really hard to dig the hole with commercial SIPRE corer (US Snow, Ice and Permafrost Research Establishment: 3" diameter). If I dig a hole with SIPRE corer for the measurement of temperature profile, the surrounding of hole will evidently be disturbed, and the area will not use it any more.

»> I expect that the soil temperature in old crustose colony will be higher than intact and old crustose sphagnum moss. Additional work will be helped me assess the difference in temperatures between new and old crustose communities through the life history.

»> R2 pointed out the prevention of degrading permafrost by the reflection of sunlight in crustose colony; however, I showed the response of air temperature to soil temperature at 2 and 5 cm depths at intact and infected colonies during the growing seasons of 2015 and 2016, as follows.

Figure B. Response from air temperature to soil temperature at 2 and 5 cm depths in intact and infected patches during the growing seasons of 2015 and 2016.

»> In Figure B, there is a little difference in gradient of temperature at 2 cm depth at intact sphagnum (green) between 2015 and 2016. However, increase of temperature gradient at 2 cm depth at crutose (blue) between both years may be the change in surface morphology at crutose infected sphagnum moss and the smooth surface at infected colony might be cracked under hot and dry growing season of 2016, as I described in chapter 3.1.

The presentation of the paper, unfortunately, is also lacking. The writing is often confusing (despite a language check) and many statements are not well-supported by either the data or a citation to another study. The authors try to solve some of the problems caused by the limited data by applying a model, but this shows a poor performance and

is subsequently extended to the full winter, a time period on which it was not tested. It's better to focus on the measurements themselves instead of using an imperfect model to draw conclusions. »> The revised text will be reviewed by the native English speaker (e.g., Nate Bauer of the International Arctic Research Center (IARC) at the University of Alaska Fairbanks) for the more understandable manuscript.

»> I checked the used data or references to cite in the text.

»> The model on the temperature sensitivity was frequently used for the flux-measuring scientists, as listed 16 cited references. As you know, growing season $CO_2$ emission elucidates > 70 % of annual carbon emission. Furthermore, Figure 8 represents the relationship between mean daily observed and simulated growing season $CO_2$ effluxes, excluding winter $CO_2$ effluxes. However, I am not sure how much contributes winter carbon emission to the annual carbon budget in crutose infected sphagnum regime despite of the significance of winter carbon emission (Natali et al., Larger loss of $CO_2$ in winter observed across the northern permafrost region, Nature Climate Change, accepted). I act a co-author in this paper.

»> I thought the readers might want the winter carbon contribution in intact and infected sphagnum.

»> Therefore, I deleted Figure 7 on simulated $CO_2$ efflux, as R2 commented. However, I want to list Figure 8 on the relationship between daily observed and simulated $CO_2$ effluxes, and Table 3 except for simulated winter $CO_2$ effluxes in the text for the assessment of temperature sensitivity.

Overall, I think it's a pity the authors did not do a better job because the data itself is truly interesting. But too many questions remain. For example: the flux measurements are only soil respiration, not net ecosystem exchange. Perhaps the growth of crustose lichen compensates for the loss of carbon from the infected Sphagnum? Unfortunately, due to the flaws in the study setup and analysis we are not closer to understanding whether crustose lichens do actually affect the $CO_2$ flux of Sphagnum mosses. »>

This study is on the investigation of soil CO2 emission (e.g., soil respiration), not net ecosystem exchange (NEE), as R2 commented. In fact, it is really difficult to examine NEE and Re in intact and infected sphagnum communities with FD chamber, not eddy covariance tower.

»> I have measured the preliminary observation (e.g., NEE and Re) with opaque and transparent chamber system in 8 intact and 8 infected sites for the growing seasons of 2013 and 2014, as previously description. Based on the investigation of NEE and Re observation, crutose lichens were completely annihilated intact sphagnum that protects the evaporation of soil moisture and the degradation of permafrost. However, the manual chamber system used in preliminary observation is constrained to monitor the continuous soil CO2 efflux-measurement, as described in L18 to L21 of page 16 (Kim et al., 2016).

»> I have wanted to investigate the difference in soil CO2 effluxes from intact and neighboring infected sphagnum after the two growing season observations, which is the aim of this study. The conclusion is increase (14%) of soil CO2 emission in crutose infected relative to intact sphagnum regime during growing seasons of 2015 and 2016.

More specific comments: Page 4, line 21-23: this statement is essential to the premise of this paper, but it's not supported by a citation. »> First of all, I corrected Otto et al. (1996) to Lange et al. (1996) in the text.

»> I corrected the L20-21 of page 4 as commented, as follows.

»> provides a protection for the photobiont as it reflects high light intensities (Ga$\beta$mann and Ott, 2000), and shows clear response characteristics with respect to light, water contents, and temperature (Hahn et al., 1993; Lange et al., 1996).

Page 4, line 27-30: this is a very basic statement but for some reason the authors need to cite 13 studies including 6 by the main author himself! One citation would suffice. »> I corrected the references in the L27-30 of page 4 as commented, as follows.

»> (Lloyd and Taylor, 1994; Davidson et al., 1998; Davidson and Janssens, 2006; Rayment and Jarvis, 2000; Oberbauer et al., 2007; Kim et al., 2007; 2013; 2014a; 2014b; 2016; Jansen et al., 2014; Kim, 2014; Euskirchen et al., 2017) »> Because it is hard to select for one representative paper to list as commented, I listed six references on the northern high latitudinal terrestrial ecosystems.

1. Temperature sensitivity: Davidson and Janssens (2006), Kim (2014), Euskirchen et al. (2017); 2. Moisture sensitivity: Oberbauer et al. (2007), Kim et al. (2014b), Jansen et al. (2014);

Page 6, lines 2-3: this paper does not specify which species of Sphagnum the measurements are done on. Judging from the photo, I assume it's Sphagnum fuscum? »> I added the name of the species to the L2-3 of page 6 as rightly commented, as follows.

»> (64°51'42.8"N; 163°42'39.1"W; 42 a.s.l.m.; Sphagnum fuscum)

Page 6, line 22: has this sensor been calibrated for moss? It's normally only calibrated for mineral soil (which Sphagnum certainly isn't). »> The commercial temperature sensor is calibrated for mineral soil excluding northern high latitude terrestrial ecosystem soil. Most of scientists have extensively used this sensor in sub-Arctic and Arctic regions.

»> Nevertheless, I did not calibrate soil temperature at the moss community. It is because 1) the depth of mineral layer is much deeper and corresponds to the top of permafrost that is the boundary (ca. 90 cm) of active layer and permafrost, and 2) the temperature in boundary layer is not representative of atmospheric temperature. Furthermore, because the sphagnum moss patches are compact, the variation of soil temperature in the patches is harmonized with change in atmospheric temperature, as previously shown in Figure B.

Page 8, line 3: this makes no sense. You estimated $CO_2$ flux sensitivity from the exponential relationship between air and soil temperature? »> I corrected the paragraph

(L2-4 of page 8) as pointed out, as follows. Actually, the relationship between air and soil temperature is not exponential, but line, as shown in Figure B.

»> We estimated the temperature sensitivity of soil $CO_2$ efflux collected by FD chamber by plotting the exponential relationship between soil $CO_2$ efflux and air temperature as well as soil temperature at depths of 2 and 5 cm, in intact and crustose lichen-infected sphagnum moss colonies, by using the following equation, as shown in Figure 5 in the text.

Page 9, line 3: air temperature at 0.5 m, but you measure at 2 m! »> I corrected the height (2.0 m) of air temperature.

Section 3: Why are the results and discussion combined? These should be separated in two sections. »> I have personally used the combination of two sections for the better understanding of readers; however, separated in '3 Results and 4 Discussion', as commented.

Page 10, line 24: what do you mean with 'forfeiture' in this context? »> It means 'loss' and I changed 'forfeiture' to 'loss'.

Page 10, line 26: indicate where this is shown in the figure. »> I added the arrows (eg, rainfall events) to Figure 2 and corrected Figure 2, as commented.

Page 11, line 2: how were these thawing rates calculated? »> I simply calculated the thawing rate with taking time between two peaks at 2 cm and 5 cm in crutose-infected and intact sphagnum moss in early spring.

Page 11, line 5: a thawing rate of 0 cm/day? »> In crutose sphagnum moss, two peaks of soil moisture at 2 cm and 5 cm depth were synchronized with same date. Then, the thawing rate between both depths is almost no difference. However, in intact moss, there is distinct difference between both depths.

Page 11, line 9-10: the sudden drop in soil moisture (and the sudden rise in spring) are probably due to the fact that your moisture sensor doesn't work below 0_ C. This is

clear from your temperature sensor. Moisture data from days with temperatures below freezing should not be used. »> I definitely agree with your comments on the fast response of soil moisture to soil temperature. However, I found that soil moisture at 2 cm of crustose moss was much higher than 5 cm, which is not common sense in 2016 despite of normal data in 2015. Exactly I am not sure the reason.

Page 11, line 16-21: these snow depth measurements appear to be from a different location, judging from the vegetation. Why not point a timelapse camera at your plots so you know when the snow melted there, rather than at another place which may not be representative of your measurement location? »> The site is within 5 m in diameter. Also, time-lapsed camera was installed the edge of the boundary. The camera may slightly move by strong wind and the pole attached to camera was heaved. Then, the photos taken by the camera seem to be different background. Also, the heavy snowfall events have frequently covered the camera and I could not measure the snow depth.

Page 11, line 11-14: it is pure speculation to say that this is due to a hotter and drier environment. Again, soil temperatures at 2 cm are lower in the crustose lichen location. Soil moisture is also regularly higher under the crustose lichen. Besides, there is no large difference in 2015 despite similar differences in moisture. »> Overall, this site is hotter and wetter growing seasons of 2015 and 2016, as described in L14-15 of page 6. I agreed with your comments, which soil temperature at 2-cm depth of crutose moss is lower than in intact sphagnum community. However, soil moisture at 2-cm depth of crutose is also lower than in intact during the growing season of 2015 (see Figure 2a), which differs from your comments. During the growing season of 2016, soil moisture at 2-cm depth of crutose is higher than in intact sphagnum since August of 2016; on the other hand, soil temperature at 2 cm of crutos is lower than in intact as 2015.

Page 14, line 8-9: this relation with soil moisture is not shown in this study. »> Yes, I did not plot the relationship soil $CO_2$ efflux and soil moisture and added the exponential equations between soil $CO_2$ efflux and soil moisture at crutose and intact sphagnum moss during the growing seasons of 2015 and 2016. I corrected and added to L8-9 of

page 14, as follows, as commented by R1 and R2.

»> CO2 flux = 2.61 exp (-8.73 × SM2 cm) (R2 = 0.08) and CO2 flux = 1.68 exp (-6.66 × SM5 cm) (R2 = 0.14) at two depths of intact sphagnum moss, and CO2 flux = 0.81 exp (-4.01 × SM2 cm) (R2 = 0.03) and CO2 flux = 6.89 exp (-13.1 × SM5 cm) (R2 = 0.12) in crustose sphagnum moss during the two growing seasons of 2015 and 2016, respectively (not shown).

Page 15, line 8: it's commonly known that air temperature governs soil temperature. There's no need to cite yourself twice to support that statement. »> I deleted two references, as commented.

Page 17, line 17: the data presented in this paper do not show a loss of ecological and thermal functions. »> I corrected L17 of P17, as follows.

»> , suggesting this may be an ecological effect of the airborne infection by crustose lichen (O. frigida) on intact sphagnum moss.

Page 17, line 27-28: by only measuring soil respiration, rather than net ecosystem exchange, it's impossible to say whether shriveled Sphagnum moss is a source of CO2 to the atmosphere. »> The monitoring of continuous soil CO2 emission with FD chamber is targeted to the soil respiration, as commented. However, I conducted CO2 flux-measurement before the installation of FD chamber for the representative site.

»> As previously described, I added the research results to the end paragraph of session 2.1.

»> The crustose sphagnum moss community cannot uptake atmospheric CO2, but is the source of ambient CO2 due to the only decomposition of dead sphagnum moss. Then I rewrote the sentence, as follows.

»> This finding demonstrates that crutose lichen-infested sphagnum moss will be a source of atmospheric CO2, and that the degradation of permafrost will be stimulated by the widespread outbreak of airborne crustose lichen on the intact sphagnum moss

community in the Subarctic and Arctic.

Please also note the supplement to this comment:
https://www.biogeosciences-discuss.net/bg-2019-121/bg-2019-121-AC2-supplement.pdf

———————————————————

[Figure]

**Fig. 1.** Temporal variations in temperature and moisture

[Figure]

[Figure]

**Fig. 2.** Temporal variations in soil CO2 emission and air temperature

[revised manuscript text omitted]

**Intact sphagnum**

**New crustose**

**Old crustose**

**Figure A. The dotted yellow line denotes the boundary of infected and intact sphagnum. Old crustose (dotted green circle) colonies represent much darker relative to new crustose (dotted green oval) and intact brown sphagnum moss communities. It represents the difference in albedo at each colony.**

**Fig. 6.** Figure A: Crustose and intact sphagnum

Figure B. Response from air temperature to soil temperature at 2 and 5 cm depths
in intact and infected patches during the growing seasons of 2015 and 2016.

**Fig. 7.** Figure B: Response from air temperature to soil temperature

---

## Author Comment (AC3) · 22 Nov 2019

The soil temperature and moisture dynamics in relation to lichen dominance should be explained a bit better. »> I have completely checked data for July-August hourly variations in soil temperature (above) and moisture (below) at intact and crustose lichen-infested sphagnum moss, as follows.

((Figure A #1)

»> However, I could not find any clues about why soil moisture at 2-cm depth of crustose lichen-infested sphagnum moss relative to 5-cm depth. Hence, we must have

[Figure]

additional observations, such as soil moisture and temperature at several centimeter depths, for the ascertainment of soil temperature and moisture dynamics for the unusual results. I appreciate your invaluable comments and suggestions for future work.

[Figure]

[Figure]

**Fig. 1.** Figure A#1

---

## Author Comment (AC4) · 24 Nov 2019

Soil CO2 efflux-measurements represent an important component for estimating an annual carbon budget in response to changes in increasing air temperature, degradation of permafrost, and snow-covered extents in the Subarctic and Arctic. However, it is not widely known the significant effect of curstose lichen (Ochrolecia frigida)-infested sphagnum moss on soil CO2 emission. Here, continuous soil CO2 efflux measurements by a forced diffusion (FD) chamber were investigated for intact and crustose lichen-infested sphagnum moss within covering over a tundra ecosystem of western Alaska during the growing seasons of 2015 and 2016. We found that CO2 efflux in

crustose lichen infested moss during the growing season of 2016 was 14 % higher than in intact sphagnum moss community. This suggests that temperature relative to soil moisture is an invaluable driver in stimulating soil $CO_2$ efflux, regardless of the restraining functions of soil moisture over emitting soil carbon. Soil moisture does not influence soil $CO_2$ emission in crustose lichen, reflecting the constraint of ecological and thermal functions relative to intact sphagnum moss. During the growing season of 2016, there was a significant difference between soil $CO_2$ effluxes in intact and crustose lichen sphagnum moss, compared to 2015, based on a t-test at the 95 % confidence level ($p < 0.05$). Mean snow-covered and snow-free $CO_2$ contributions to annual carbon budgets correspond to 28.4 % and 71.6 % in intact sphagnum moss, and 25.0 % and 75.0 % in a crustose lichen sphagnum moss, respectively. There-fore our findings demonstrate that soil $CO_2$ emissions in the crustose lichen-infested sphagnum moss would be steadily stimulated by a widespread outbreak of airborne lichen over intact sphagnum moss. This might result in rapid degradation of permafrost in the Subarctic and Arctic.